# A dual interaction between RSV NS1 and MED25 ACID domain reshapes antiviral responses

Celia Ait-Mouhoub[1☉], Jiawei Dong[2☉], Magali Noiray[3], Jenna Fix[1], Stepanka Nedvedova[2], Slim Fourati[4], Alexis Verger[5], Jean-Francois Eleouet[1], Delphyne Descamps[1], Monika Bajorek [1]*, Christina Sizun[2]*

1 Virologie et Immunologie Moléculaire, INRAE, UVSQ, Université Paris-Saclay, Jouy-en-Josas, France,
2 Institut de Chimie des Substances Naturelles, CNRS, Université Paris-Saclay, Gif-sur-Yvette, France,
3 Institute for Integrative Biology of the Cell, CNRS, CEA, Université Paris-Saclay, Gif-sur-Yvette, France,
4 Department of Virology, Hôpitaux Universitaires Henri Mondor, AP-HP, Université Paris-Est-Créteil, INSERM, Créteil, France, 5 EMR 9002 Integrative Structural Biology, CNRS, INSERM, Université de Lille, Institut Pasteur de Lille, Lille, France

☉ These authors contributed equally to this work.
* christina.sizun@cnrs.fr (CS); monika.bajorek@inrae.fr (MB)

## Abstract

Respiratory syncytial virus (RSV), the most common cause of bronchiolitis and pneumonia in infants, elicits a remarkably weak innate immune response. This is partly due to type I interferon (IFN) antagonism by the non-structural RSV NS1 protein. It was recently suggested that NS1 could modulate host transcription via an interaction with the MED25 subunit of the Mediator complex. Previous work emphasized the role of the NS1 C-terminal helix α3 for recruitment of the MED25 ACID domain, a target of transcription factors (TFs). Here we show that the NS1 α/β core domain binds to MED25 ACID and acts cooperatively with NS1 α3 to achieve nanomolar affinity. The strong interaction is rationalized by the dual NS1 binding site on MED25 ACID predicted by AlphaFold and confirmed by NMR, which overlaps with the two canonical binding interfaces of TF transactivation domains. Single amino acid substitutions in the NS1 α/β domain, notably NS1 E110A, significantly reduced the affinity of NS1 for MED25 ACID, both in vitro and in cellula. These mutations resulted in attenuated replication of recombinant RSV (rRSV-mCherry). They did not significantly upregulate type I or III IFN levels in IFN-competent BEAS-2B cells, contrary to the NS1 α3 deletion. However, in line with attenuated replication, the NS1 E110A mutation enhanced expression of the antiviral interferon-stimulated gene ISG15, and NS1 I54A upregulated ISG15, OAS1A and IFIT1 in IFN-competent cells. In MED25-knockdown cells, rRSV-mCherry replication was further attenuated at a late post-infection timepoint. The difference between WT and NS1 mutant rRSV-mCherry was partially lost, suggesting that the NS1–MED25 ACID complex contributes to controlling antiviral responses at this timepoint. The strong interaction and the extended binding interface between NS1 and MED25 ACID provide evidence for a mechanism, where

**Data availability statement:** All relevant data are within the manuscript and its Supporting information files.

**Funding:** The present work has benefited from the I2BC-PIM platform supported by the French Infrastructure for Integrated Structural Biology FRISBI (grant number ANR-10-INBS-05), https://frisbi.eu/. J.D. was supported by a doctoral grant of Université Paris-Saclay, Doctoral School SDSV, https://www.universite-paris-sa-clay.fr/ecoles-doctorales/innovation-therapeu-tique-du-fondamental-lapplique-itfa. C.A.-M. was supported by doctoral grants of Université Paris-Saclay, Doctoral School ABIES, https://abies.agroparistech.fr/, and of the INRAE Animal Health Department, https://www.inrae.fr/departements/sa. M.B. and C.S. acknowledge financial support (grant number DIMIHEALTH 2021-12) by DIM One Health 2.0, https://www.dim1health.com/. The funders had no role in study design, data collection and analysis, decision to publish, or preparation of the manuscript.

**Competing interests:** The authors have declared that no competing interests exist.

NS1 blocks access of transcription factors to MED25, and thereby MED25-mediated transcription activation.

---

## Author summary

Respiratory syncytial virus (RSV) is a major pathogen for acute lower respiratory infections in infants and in the elderly. RSV elicits a remarkably weak immune response. It has developed a unique strategy to counteract the immune system, by encoding two small multifunctional proteins, RSV NS1 and NS2. NS1 is involved in interferon antagonism in the cytosol. Recently NS1 was shown to modulate host transcription in the nucleus. However, the mechanisms underpinning this function are not fully clear. Here we focus on the interplay between NS1 and the cellular MED25 coactivator protein, which can contribute to the antiviral response by activating innate immune response genes. The MED25 C-terminal ACtivator Interacting Domain (ACID), a target of cellular transcription factors (TF), is a key feature for this function. To investigate the impact of MED25 hijacking by NS1, we combined in vitro biophysical experiments and cellular assays to probe the relationship between the stability of the NS1–MED25 ACID complex and RSV replication as well as antiviral responses. Our results suggest that this interaction is correlated with antiviral response antagonism, probably by hindering TFs to interact with MED25 ACID. This knowledge might pave the way for antiviral strategies aimed at stimulating appropriate immune responses.

## Introduction

Human respiratory syncytial virus (hRSV) is the most common cause of acute lower respiratory tract infections in infants and elderly [1]. In 2019, an estimated 33 million cases of RSV occurred in infants worldwide, requiring 3.6 million hospitalizations and resulting in ~100,000 deaths among children under the age of five [2]. While vaccines for older adults and pregnant women, as well as prophylactic antibodies for infants, were marketed recently, there is still no affordable effective specific treatment for RSV [3,4]. RSV elicits a remarkably weak innate immune response, and RSV-infected infants produce very low type I interferon (IFN-I) levels [5]. To find new directions for antiviral therapies, a better understanding of RSV pathogenesis, in particular of RSV immune response evasion, is needed.

RSV is an enveloped, negative-strand RNA virus of the *Pneumoviridae* family, *Mononegavirales* order [6]. Its genome is composed of 10 genes encoding 11 proteins, including two non-structural proteins, NS1 and NS2. RSV NS1 and NS2 have no homologs outside the *Orthopneumovirus* genus. They constitute a unique strategy of RSV to evade the host immune system. NS1 and NS2 interfere with several IFN induction mechanisms and act individually or together as antagonists of antiviral pathways [7–11].

The RSV NS1 gene is the first transcribed, suggesting a critical role at the beginning of infection. NS1 is a small protein of 139 amino acids that interferes with IFN-I induction and signaling [10,11]. Several cytoplasmic targets of NS1 have been identified. NS1 interacts with E3-ubiquitin ligase TRIM25 and inhibits the ubiquitination of innate immune receptor RIG-I, thereby preventing RIG-I mediated IFN production [12]. NS1 acts as a scaffold protein for a multi-subunit E3 ligase complex, leading to degradation of TNF receptor associated factor 3 (TRAF3), thereby lowering induction of interferon stimulated genes (ISGs) [13–15]. It was recently reported that NS1 interacts with interferon stimulated BST2/Tetherin [16]. BST2 was hypothesized to be a restriction factor against RSV acting at the budding stage. NS1 also prevents premature apoptosis [17]. Furthermore, NS1 was hypothesized to contribute to host-range restriction [18].

A fraction of RSV NS1 is present in the nucleus of infected epithelial cells, where it associates with host nuclear proteins [15,19,20]. It was recently shown that RSV NS1 modulates host cell gene transcription via interactions with chromatin and with the Mediator complex [21]. The multi-subunit Mediator complex is a transcriptional co-regulator that is essential for the regulation of RNA polymerase II transcription in eukaryotes [22–24]. Its main function is to transmit regulatory information from sequence specific transcription factors (TFs) bound to enhancer regions to the promoter-bound basal RNA pol II machinery. The Mediator is hijacked by a number of viruses, in particular DNA viruses and retroviruses, e.g., herpes simplex virus, human papilloma virus and Walleye dermal sarcoma virus, to control viral and host gene expression [25]. The Mediator subunit MED25 was first found to be an interactor of NS1 by quantitative proteomics [19]. This interaction was independently confirmed by affinity purification-mass spectrometry [21], a yeast two-hybrid screen and luciferase complementation [26], and a BioID proximity screen complemented by two other binary protein-protein interaction screens [27]. Finally, RSV replication was shown to be enhanced in interferon-competent epithelial A549 lung cells, where MED25 was knocked out [27], establishing a possible link between NS1 interferon antagonism and MED25 regulated transcription.

MED25 contains two folded domains connected by a long disordered loop and a disordered Q-rich C-terminal region (**Fig 1A**). The N-terminal von Willebrand (VWA) domain anchors MED25 to the Mediator. The C-terminal ACtivator Interacting Domain (ACID) of MED25 is recruited by acidic transactivation domains (TADs) of DNA-binding TFs [28]. Its structure was solved by NMR, and contains a β-barrel domain flanked by three α-helices and dynamic loops [29–32].

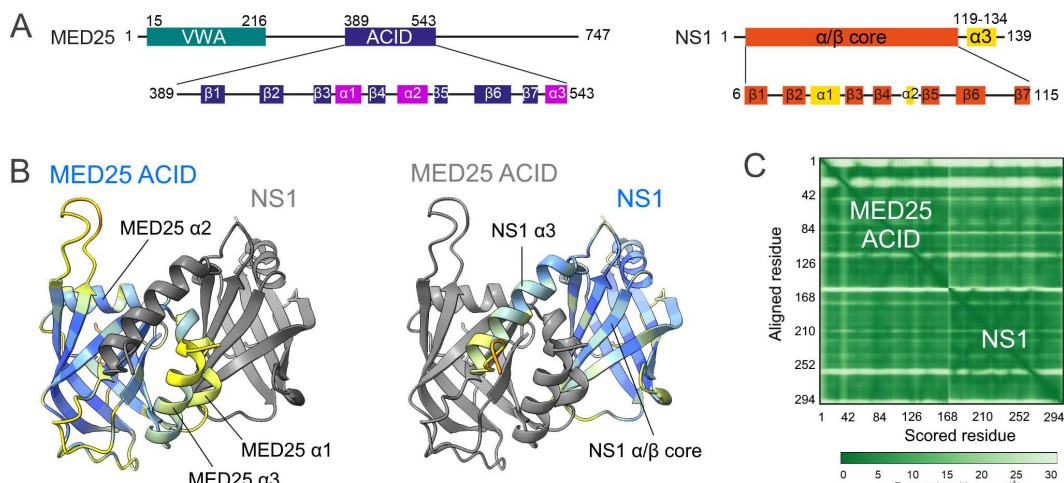

**Fig 1. AlphaFold (AF) structural prediction of the complex between human MED25 ACID and RSV NS1. (A)** Line diagram of MED25, highlighting the von Willebrand A (VWA) and ACtivator Interacting (ACID) domains, and secondary structures of MED25 ACID and RSV NS1. **(B)** Highest ranked AlphaFold model of the NS1–MED25 ACID complex. Proteins are in cartoon representation. MED25 ACID is displayed with the α3 helix and the H1 interface in front view. The positions of α-helices of both proteins and of the NS1 α/β core domain are marked. Individual protein models were colored either in gray or according to the predicted local-distance difference test (pLDDT) confidence score, using the ChimeraX AlphaFold color palette [43]. (C) Predicted aligned error (PAE) matrix for the complex with residues 1-155 (MED25 ACID) and residues 156-294 (NS1).

Mammalian MED25 ACID displays two distinct TAD binding interfaces, termed H1 and H2, which are located on two opposite faces of the β-barrel. ETV/PEA3 family TADs target the H1 interface [33,34], whereas ATF6α targets the H2 interface [35,36]. Tandem TADs of p53 [37] and *Herpes simplex* VP16 each bind to one of these interfaces [28]. We previously identified MED25 ACID as the binding domain of NS1 [26].

RSV NS1 contains a compact α/β core domain, composed of a 7-stranded β-sandwich domain and α-helix α1, and a C-terminal α-helix α3 (**Fig 1A**) [38]. NS1 α3 was reported to be involved in the modulation of host response and gene expression [21,38]. We showed that the NS1 α3 deletion strongly impaired MED25 ACID binding, indicating that α3 is a main determinant for the formation of the NS1–MED25 ACID complex [26]. NMR experiments indicated that NS1 α3, as an isolated peptide, bound to the MED25 ACID H2 interface, similarly to H2 binding TADs [26]. Moreover, NS1 was able to compete for binding with the TAD of ATF6α, a master regulator of the endoplasmic reticulum stress response, which activates the unfolded protein response (UPR) [39]. ATF6α TAD also binds to the MED25 ACID H2 interface [36]. Unlike DNA-specific TFs that recruit MED25, NS1 lacks a DNA binding domain, and hence cannot directly transduce information to a promoter. Due to the lack of DNA-binding motifs in many viral transcription regulators, it has been suggested that these regulators engage in protein-protein interactions with host regulators rather than directly bind to nucleic acids [40]. We thus hypothesized that NS1 may impair host gene transcription activation by occluding the H2 TAD binding site of MED25 ACID and preventing its recruitment by cellular TFs [26].

However, many aspects of the NS1−MED25 ACID interaction remained puzzling. In particular, the NS1 α3 peptide displayed only 10 μM affinity, as compared to 20 nM for full-length NS1 [26]. We therefore reasoned that the NS1 α/β core domain also binds to MED25 ACID, and found out that NS1 α3 and the α/β core domain cooperatively bind to MED25 ACID. Ab initio prediction of the MED25 ACID–NS1 complex structure suggested that NS1 binds to both H1 and H2 interfaces via its α3 and α/β core subdomains, respectively. Alanine substitution of NS1 α/β core domain residues located in the interface reduced the NS1−MED25 ACID interaction, and some single amino acid mutations were as potent as the deletion of NS1 α3. These mutations attenuated replication of recombinant rRSV-mCherry, but did not significantly upregulate IFN-I or IFN-III levels in IFN-competent cells. However, two of the NS1 α/β core domain mutations, I54A and E110A, induced higher transcription levels for antiviral ISGs, in particular ISG15. In A549 cells depleted of MED25, we observed that replication was attenuated for WT rRSV-mCherry as well as for NS1 mutants at a late timepoint post-infection. Moreover, at this timepoint, the difference between WT rRSV-mCherry and NS1 mutants was less marked than in control cells, indicating that the NS1–MED25 ACID complex may come into play at this stage. Based on these results, we hypothesize that RSV uses MED25 to evade cellular antiviral responses by precluding activation via MED25-specific TFs, due to efficient competitive binding of the entire NS1 to both TAD binding sites of MED25 ACID.

## Results

### Ab initio structural model of the NS1–MED25 ACID complex

We previously observed by NMR that the isolated NS1 α3 peptide bound to the MED25 ACID H2 interface, with a Kd of 10 μM [26]. NS1 α3 induced NMR signal perturbations similar to those of H2-binding TADs [29,32]. We also observed secondary binding to the H1 interface by NMR, but with much lower affinity (Kd 500 μM). Structure prediction of the NS1 α3–MED25 ACID complex with machine-learning algorithms suggested that NS1 α3 binds to the canonical TAD-binding interfaces of MED25 ACID, but could not discriminate between H1 and H2 interfaces [28]. Moreover, we had previously noticed that deletion of α3 not fully impaired MED25 ACID binding [26]. Therefore, we generated ab initio structural complex models by deep learning methods using full-length NS1. High accuracy models were obtained with AlphaFold2 [41] and AlphaFold3 [42]. The two algorithms provided similar results (**Fig 1B**).

The individual protein structures were rather well predicted, with predicted template modeling (pTM) scores of 0.75 and 0.79, for MED25 ACID and NS1, respectively. The two MED25 ACID helices α1 and α3 were predicted with lower

predicted local-distance difference test (pLDDT) scores (**Fig 1B**). The large loops of MED25 ACID were also less well defined. This is consistent with previous reports that described the latter as highly dynamic [36,44]. A high confidence was found for the complex interface with an interface pTM (ipTM) score of 0.72, as illustrated by the predicted aligned error (PAE) matrix (**Fig 1C**). In all complex models, NS1 was clamped onto the MED25 ACID β-barrel on each side of the α1/α3 helix pair. Surprisingly, the C-terminal NS1 α3 helix was predicted to bind to the H1 interface, while the NS1 α/β core made extensive contacts with the H2 interface. Notably, α3 is no longer associated to the core domain, unlike in the dimeric NS1 crystal structure [38], where α3 contributes to intra- and inter-protomer contacts. This suggests that potential dimers are disrupted upon complex formation. This hypothesis is supported by the same NS1 β-sheet being involved in the NS1 dimer and in the MED25 ACID interface.

While this paper was under review, an X-ray crystal structure of the NS1–MED25 ACID complex was published (PDB 9ccv) [45]. Our AlphaFold model and the experimental structure superimpose very well, which emphasizes the high quality of the prediction (**S1 Fig**). Of note, NS1 residue D102 in our construct is mutated, but it is outside the interaction region. This residue is N102 in RSV strains Long and A2. In summary, AlphaFold predicts that full-length NS1 uses a dual binding site on MED25 ACID, comprising both TAD-binding interfaces. It also suggests that conformational changes occur upon complex formation both in NS1 and MED25 ACID, presumably in the less well defined protein regions.

## Binding of NS1 with α3 deletion to MED25 ACID

To assess the contribution of the NS1 α/β core domain to MED25 ACID binding, we measured real time association/dissociation kinetics using bio-layer interferometry (BLI). We used recombinant His-tagged MED25 ACID as a ligand, and full-length NS1 or the C-terminally truncated NS1Δα3 (NS1 aa 1–116) proteins as analytes. Measurements were done in Tris pH 8, NaCl buffer at a temperature of 25°C. Under these conditions, NS1 does not self-assemble [46]. His-tagged MED25 ACID was successfully loaded onto Ni-NTA biosensors, which were subsequently incubated with NS1 or NS1Δα3. The NS1 BLI data were well fitted with a 1:1 binding model (**Table 1** and **S2 Fig**). A Kd of 16 nM was determined, in line with previous ITC data (15 nM). Association of NS1 was rather slow with a rate of 0.12 $\mu M^{-1}.s^{-1}$, as compared to the values reported for the TADs of VP16, ERM or ATF6α (300–1,100 $\mu M^{-1}.s^{-1}$) [36]. Dissociation was also slow, with a rate of 2 $10^{-3}$ $s^{-1}$ versus 100–400 $s^{-1}$ for these TADs.

Contrary to NS1, a single binding site model failed to reproduce BLI data obtained with NS1Δα3. However, the data were well fitted with a 2:1 heterogeneous binding model (**S2 Fig** and **Table 1**). This model applies in the case, where two different populations of NS1–MED25 ACID complex are formed, both with 1:1 stoichiometry. The two binding modes of NS1Δα3 afforded close Kd values (615 and 850 nM), confirming that the NS1 α/β core domain binds to MED25 ACID on its own.

While the two binding modes of NS1Δα3 cannot be distinguished in terms of affinity, they differ in terms of kinetics. Mode 1 (75% populated) displayed both faster association and faster dissociation than mode 2 (25% populated) (**Table 1**). Overall, the association rates for NS1Δα3 were lower than for full-length NS1, which points to the contribution of NS1 α3. In binding mode 1, the association rates of NS1Δα3 and NS1 were comparable, but NS1Δα3 dissociated faster, resulting

**Table 1. Bio-layer interferometry (BLI) binding equilibrium and kinetic parameters for the NS1 and NS1Δα3 complexes with MED25 ACID.**

| | Kd$_1$ (nM) | Kd$_2$ (nM) | % mode 1 | % mode 2 | ka$_1$ ($\mu M^{-1}.s^{-1}$) | ka$_2$ ($\mu M^{-1}.s^{-1}$) | kdis$_1$ ($10^{-3}$ $s^{-1}$) | kdis$_2$ ($10^{-3}$ $s^{-1}$) |
|---|---|---|---|---|---|---|---|---|
| NS1 | 16.5±0.04 | / | 100% | / | 0.12 ± 0.0002 | / | 2.0 ± 0.003 | / |
| NS1Δα3 | 615±5 | 850±8 | 75% | 25% | 0.08 ± 0.0006 | 0.003 ± 3 $10^{-5}$ | 50 ± 0.1 | 2.4 ± 0.009 |

Kd dissociation constant (nM), ka association rate ($\mu M^{-1}.s^{-1}$), kdis dissociation rate ($10^{-3}$ $s^{-1}$). BLI data at pH 8 and 25°C were fitted with a 1:1 binding model for NS1, and with a 2:1 heterogeneous binding model for NS1Δα3.

in lower affinity, and suggesting the formation of an encounter complex. Conversely, in mode 2, the dissociation rates were similar for NS1Δα3 and NS1, but NS1Δα3 associated more slowly, resulting again in a higher Kd. This second complex may result from slow rearrangements taking place after formation of the mode 1 complex, following an induced fit mechanism. However, it cannot be excluded that it results from conformational selection of either NS1 Δα3 or MED25 ACID, assuming conformational equilibria for these proteins.

The affinity of NS1Δα3 is 1–2 orders of magnitude lower than that of full-length NS1, and the affinity of the NS1 α3 peptide (10 μM, measured by NMR and by ITC [26]) is 3 orders of magnitude lower. The substantial affinity gain of full-length NS1 versus its isolated α3 and α/β core subdomains strongly indicates cooperative binding. This can only be rationalized by a dual binding mode, as proposed in the AlphaFold model of the NS1–MED25 ACID complex (**Fig 1B**). To rule out a bias due to an altered protein structure, we measured thermal protein unfolding using dynamic scanning fluorimetry (DSF). NS1 and NS1Δα3 displayed similar sample brightness and 350 nm/330 nm fluorescence ratios. Both proteins showed cooperative unfolding, with transition temperatures of 69.5°C and 65.8°C, respectively (**S3 Fig**). This indicates that although the Δα3 mutation slightly destabilizes NS1, NS1Δα3 is folded similarly to the α/β core domain in NS1.

To get structural insight, we generated AlphaFold models of the NS1Δα3–MED25 ACID complex. The interface confidence was lower than with NS1 (ipTM 0.6). Nevertheless, the models indicate that NS1Δα3 targets the H2 interface of MED25 ACID. In the absence of α3, NS1Δα3 binds to MED25 ACID similarly to the α/β domain of NS1, with only small rearrangements (**S4A Fig**). The NS1 α/β core domain is slightly rotated on the H2 interface, and the two MED25 ACID α1 and α3 helices are slightly displaced with respect to the MED25 ACID β-barrel (**S4B Fig**). This suggests that conformational changes may occur in MED25 ACID α1 and α3 helices, and that these helices may couple the two H1 and H2 interfaces.

To obtain experimental evidence for binding of the NS1 α/β core domain to the H2 interface of MED25 ACID, we used NMR. We measured 2D $^1$H-$^{15}$N HSQC spectra of $^{15}$N-labeled MED25 ACID mixed with unlabeled NS1Δα3 or NS1. Samples were prepared in a buffer at pH 6.5 to ensure that amide signals were not broadened by solvent exchange. At a 1:1 molar ratio, most MED25 ACID amide NMR signals became nearly undetectable due to severe line broadening. This is indicative of the formation of a protein complex, as a size increase leads to more efficient transverse $^{15}$N relaxation. To detect differential line broadening induced by chemical exchange at the interface of the complex, we measured intensities at a lower NS1Δα3 or NS1 molar ratio (**Fig 2A** and **2B**). At a 0.5 ratio, NS1Δα3 and NS1 induced similar perturbation patterns. Global line broadening was observed, and residual intensities I/I0 were ~0.5. Only sharp signals, clustered at the center of the HSQC spectrum, which also displayed higher intensities in free MED25 ACID, displayed higher I/I0 values (areas shaded in grey in **Fig 2C** and **2D**). They were assigned to the disordered N-terminus (A373-M388) and to a long flexible loop (E410-L423), which remain flexible in the complex. Line broadening for the ordered regions of MED25 ACID was not uniform. This was more marked for NS1Δα3 compared to NS1, and may be explained by the faster binding kinetics. In particular, MED25 ACID residues 450–470 (helix α1 and strand β4) and residues 510–542 (strands β6 and β7), encompassing the H2 interface of MED25 ACID, displayed lower intensities, and delineated a large perturbation area, when mapped onto the 3D structure of MED25 ACID (**Fig 2E** and **2F**). Perturbations observed for the C-terminal α3 helix of MED25 ACID, which is not part of the H2 interface, but antiparallel to the α1 helix, suggest that α3 might sense conformational changes in α1 as a result of NS1Δα3 or NS1 binding. Taken together, our NMR data indicate that the NS1 α/β core domain targets the H2 interface of MED25 ACID, in agreement with the AlphaFold models, and in apparent contradiction with previous NMR data, where NS1 α3 targeted the same interface.

## Identification of NS1 α/β core domain residues critical for MED25 ACID binding in vitro

Based on the NS1–MED25 ACID AlphaFold prediction, we individually mutated five NS1 interface residues into alanine. M122 is at the N-terminus of helix α3. The four other substituted amino acids are located on the B1 β-sheet of NS1: I54 and F56 on strand β4, and E110 and K112 on strand β7 (**Fig 3A**). I54, F56, and E110 were chosen close to the MED25

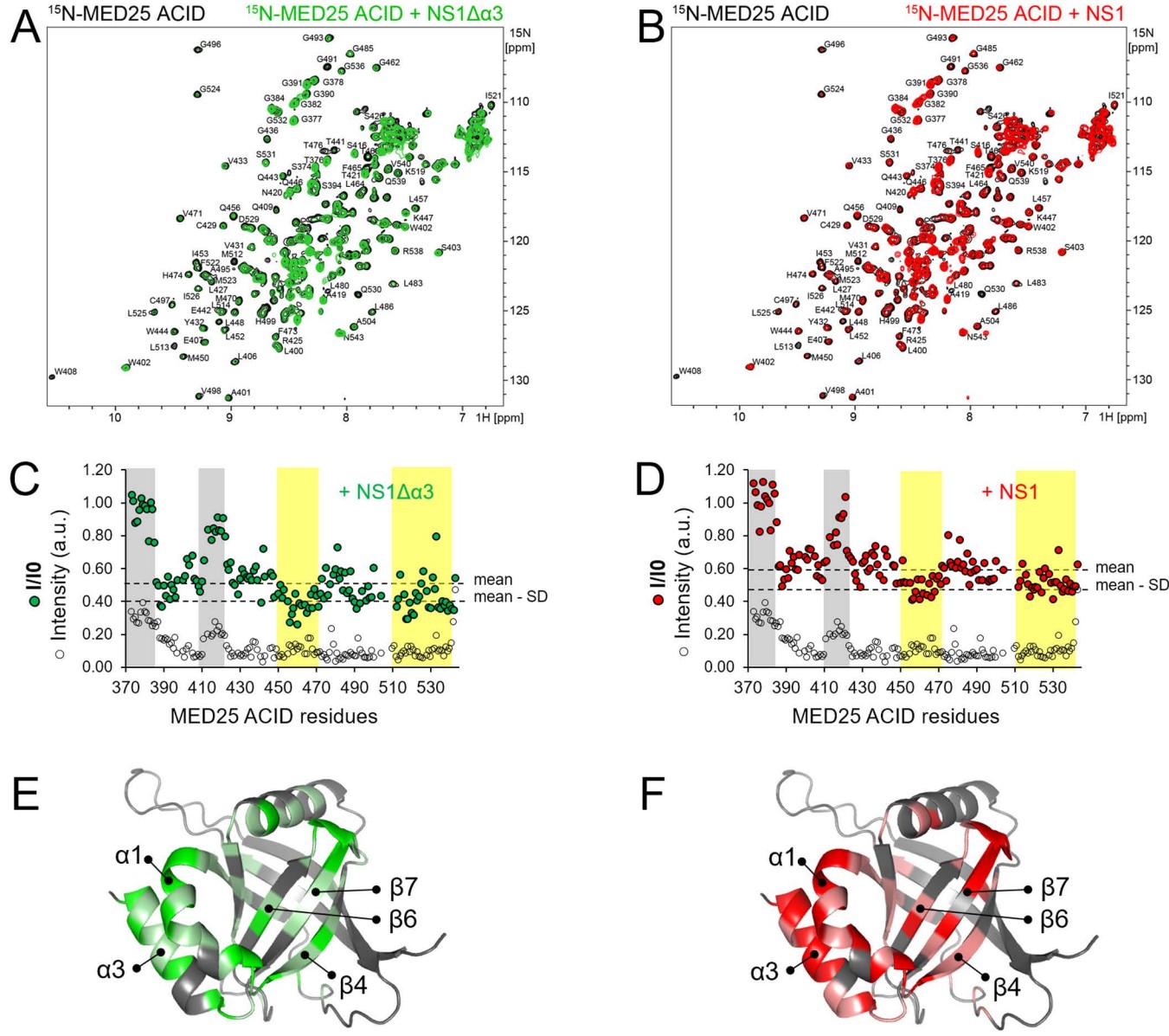

**Fig 2. NMR analysis of NS1 and NS1Δα3 binding to MED25 ACID. (A,B)** Superimposition of 2D $^1$H-$^{15}$N HSQC spectra of $^{15}$N-MED25 ACID (100 μM) alone (black) and mixed with 0.5 molar equivalent of NS1Δα3 (green, **A**) or full-length NS1 (red, **B**). Samples were prepared in 20 mM Na phosphate pH 6.5, 100 mM NaCl, 5 mM TCEP buffer. Experiments were carried out at 800 MHz $^1$H resonance frequency and at a temperature of 20°C. **(C,D)** Residual intensities of MED25 ACID were calculated as the ratios of intensities in the presence and absence of NS1Δα3 or NS1 for each amide signal (I/I0, filled dots). The mean and mean−SD values were drawn with broken lines. Mean values and standard deviations (SD) were calculated from residues in folded regions. The amide signal intensities of MED25 ACID alone were plotted on the same diagram with arbitrary units (empty dots). Disordered regions at the N-terminus of MED25 ACID and in the E410-L423 loop were highlighted in grey. Regions with I/I0 < mean−SD were highlighted in yellow. **(E,F)** Residues with marked intensity attenuation were mapped onto the 3D structure of MED25 ACID represented in cartoon. The H2 TAD-binding interface is in front view. Residues with mean−SD < I/I0 < mean are in light color and I/I0 < mean−SD in bright color.

ACID M523 residue (**Fig 3A**), as the M523E mutation disrupted the interaction with NS1 [26]. An NCBI Virus blast query showed that these five positions are highly conserved within a set of 471 human RSV NS1 sequences: 16 mismatches were detected for I54, 9 for F56, 2 for E110, 2 for K112, and 1 mismatch for M122.

We produced recombinant mutated NS1 proteins, and measured the interaction with MED25 ACID by BLI. All NS1 variants made specific interactions with MED25 ACID. Similarly to NS1Δα3, data could not be accurately fitted with a single site binding model. We thus extracted thermodynamic and kinetic parameters with a 2:1 heterogeneous binding model (**S2 Fig**). The affinities spanned a range over two orders of magnitude with Kd values between 7.5 nM and 1.6 µM (Table 2 and Fig 3B). NS1 single amino acid mutations E110A (Kd1 1.3 µM, Kd2 1.6 µM) and F56A (Kd1 330 nM; 80% populated) were the most efficient to disrupt the complex, and the loss of affinity was comparable to that of the NS1 Δα3 deletion.

All mutated NS1 proteins displayed a binding mode with faster association (mode 1). The association rates ($ka_1$ 0.12-0.21 µM⁻¹.s⁻¹) were close to that of NS1 (0.12 µM⁻¹.s⁻¹), and slightly higher than that of NS1Δα3 (0.08 µM⁻¹.s⁻¹) (**Table 2 and Fig 3C**). Mode 1 was the lower affinity mode, and the strength of the interaction inversely correlated with the dissociation rate ($kdis_1$ 9–190 $10^{-3}$ s⁻¹). Mode 1 was higher populated for NS1 with single amino substitutions that were most impaired for MED25 ACID binding (F56A, E110A). The higher affinity binding mode (mode 2) generally displayed slower

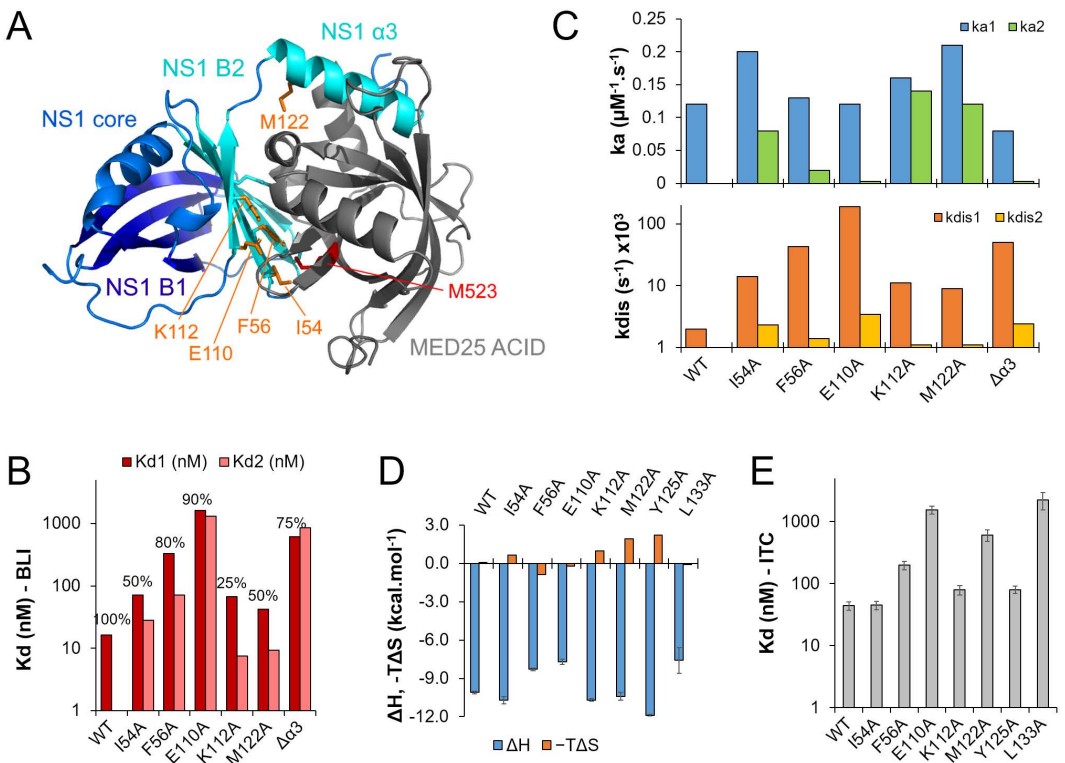

**Fig 3. Mutational analysis of NS1 binding to MED25 ACID in vitro. (A)** Residues selected for mutational analysis, located at the interface between NS1 and MED25 ACID, based on the AlphaFold NS1–MED25 ACID complex structure. Proteins are in cartoon representation. MED25 ACID is in gray, and NS1 in blue and cyan. The NS1 β-sheet B1 is in dark blue, β-sheet B2 and helix α3 in cyan. NS1 residues that were mutated into alanine (I54, F56, E110, K112, and M122) are in orange sticks. MED25 ACID M523, which was previously shown to be involved in the NS1–MED25 ACID interaction, is in red sticks. **(B)** Dissociation constants (Kd) of the NS1–MED25 ACID complex measured by BLI for WT and mutated NS1. A 2:1 heterogeneous binding model was used for mutated NS1. The relative population (%) of the first binding mode is indicated on top of the bars. **(C)** Association (ka) and dissociation (kdis) rates measured by BLI. **(D)** Enthalpy (ΔH) and entropic (-TΔS) contributions to free energy measured by Isothermal Titration Calorimetry (ITC) for MED25 ACID binding to WT and mutated NS1. **(E)** Dissociation constants (Kd) measured by ITC.

**Table 2. Bio-layer interferometry (BLI) binding equilibrium and kinetic parameters for the complex of NS1 with single amino acid substitutions with MED25 ACID.**

| | Kd1 (nM) | Kd2 (nM) | % mode 1 | % mode 2 | $ka_1$ (µM$^{-1}$.s$^{-1}$) | $ka_2$ (µM$^{-1}$.s$^{-1}$) | $kdis_1$ ($10^{-3}$ s$^{-1}$) | $kdis_2$ ($10^{-3}$ s$^{-1}$) |
|---|---|---|---|---|---|---|---|---|
| NS1 I54A | 71.2±0.5 | 28.3±0.2 | 50% | 50% | 0.20 ± 0.001 | 0.08 ± 0.0003 | 14 ± 0.05 | 2.3 ± 0.007 |
| NS1 F56A | 333±1 | 72.0±0.6 | 80% | 20% | 0.13 ± 0.0005 | 0.02 ± 0.0001 | 43 ± 0.07 | 1.4 ± 0.006 |
| NS1 E110A | 1 630±20 | 1 320±20 | 90% | 10% | 0.12 ± 0.001 | 0.003 ± 4 $10^{-5}$ | 190 ± 0.9 | 3.4 ± 0.02 |
| NS1 K112A | 68.1±1.3 | 7.5±0.1 | 25% | 75% | 0.16 ± 0.003 | 0.14 ± 0.0006 | 11 ± 0.06 | 1.1 ± 0.003 |
| NS1 M122A | 42.3±0.3 | 9.4±0.05 | 50% | 50% | 0.21 ± 0.001 | 0.12 ± 0.0003 | 9 ± 0.03 | 1.1 ± 0.004 |

Kd dissociation constant (nM), ka association rate (µM$^{-1}$s$^{-1}$), kdis dissociation rate ($10^{-3}$ s$^{-1}$). Data were measured at pH 8 and 25°C and fitted with a 2:1 heterogeneous binding model.

association. It also displayed slower dissociation, with dissociation rates ($kdis_2$ 1.1-3.4 $10^{-3}$ s$^{-1}$) in the same range as that of WT NS1 (2 $10^{-3}$ s$^{-1}$). Mode 2 was significantly populated for the mutations that were less efficient to disrupt disrupted MED25 ACID binding, with 75% population for the K112A mutation.

To validate the binding parameters with a second method, we made measurements by isothermal titration calorimetry (ITC), by titrating MED25 ACID into WT or mutated NS1. The experimental conditions were optimized to avoid aggregation of NS1. Surprisingly, no heat was measured for NS1Δα3. But a similar result was recently reported [45]. This might point to conformational rearrangements concomitant to NS1Δα3–MED25 ACID complex formation. However, the five NS1 proteins with alanine substitutions were amenable to ITC (S5 Fig). The ITC data could be fitted with a single binding site model, resulting in a global thermal signature at equilibrium. Binding was mainly enthalpy driven (Table 3 and Fig 3D). The Kd values (Table 3 and Fig 3E) were of the same order of magnitude as the Kds measured by BLI, except for M122A. These data confirm the role of residues F56 and E110 for MED25 ACID binding. To rule out a possible bias due to structural integrity, we verified the stability of mutated NS1 by thermal denaturation. The DSF curves and F350/F330 ratios were similar to that of WT NS1. Inflection temperatures varied between 65.6°C and

**Table 3. Thermodynamic parameters for binding of wild-type NS1 and NS1 with single amino acid substitutions to MED25 ACID measured by ITC.**

| | N | Kd (nM) | ΔH (kcal.mol$^{-1}$) | -TΔS (kcal.mol$^{-1}$) |
|---|---|---|---|---|
| NS1-WT | 0.73 | 44±7 | -10.1±0.1 | 0.05 |
| NS1-I54A | 0.79 | 45±7 | -10.7±0.3 | 0.64 |
| NS1-F56A | 0.80 | 198±31 | -8.3±0.1 | -0.89 |
| NS1-E110A | 0.86 | 1540±230 | -7.7±0.2 | -0.24 |
| NS1-K112A | 0.66 | 79±13 | -10.7±0.1 | 0.96 |
| NS1-M122A | 0.71 | 606±125 | -10.4±0.3 | 1.9 |
| NS1-Y125A | 0.53 | 80±11 | -11.9±0.1 | 2.2 |
| NS1-L133A | 0.44 | 2230±700 | -7.6±1.0 | -0.08 |

Number of sites (N), dissociation constant (Kd), association enthalpy (ΔH), and entropic contribution to free energy (-TΔS). Measurements were carried out at pH 8 and at 25°C. ITC data were fitted with a one set of sites model.

69.3°C, confirming that these proteins were all folded (S3 Fig). Thus, no correlation was observed between the stability of mutated NS1 and its binding strength. To compare with previously investigated NS1 α3 residues [26,38], we measured ITC data for the two NS1 single amino acid mutations Y125A and L133A. The NS1 Y125A mutation only moderately impaired MED25 ACID binding, in agreement with our previous interaction assays in cells [26]. In contrast, the L133A mutation efficiently disrupted the complex with a Kd of ~2 µM (Table 3 and Fig 3E). This is in line with previous findings, where the double L132A/L133A substitution in NS1 disrupted MED25 ACID binding in cells [26], and suggests that L133, rather than L132, is a crucial residue for the NS1–MED25 ACID complex. Taken together, our results indicate that the NS1 α/β core domain plays a critical role for MED25 ACID binding, and that single amino acid mutations in both NS1 core and α3 domains can destabilize the complex with a loss of affinity by up to nearly two orders of magnitude, strengthening confidence in the AlphaFold model.

### Validation of NS1 α/β core domain binding to MED25 ACID in cellula

To validate the NS1 mutations in cells, we used a split-luciferase complementation assay, based on the NanoLuc enzyme [47]. The 114 NanoLuc subunit was fused to the C-terminus of WT or mutated NS1 (NS1-114), and the 11S subunit to the N-terminus of MED25 ACID (11S-MED25 ACID). The constructs were co-transfected into HEK 293T cells. Controls were made by co-transfecting each fusion construct with its complementary non-fused NanoLuc construct. Cells were lysed 24 h post-transfection, and luciferase substrate was added. The interaction was measured using the luminescence signal. When NS1-114 was co-expressed with 11S-MED25 ACID, the luminescence was high (**Fig 4A**), indicating that the NS1–MED25 ACID complex was formed. The NS1Δα3–MED25 ACID interaction was used as a control. The α3 deletion drastically reduced the interaction with MED25 ACID, in line with the significant loss of affinity observed with NS1Δα3 in vitro. We then measured interactions using the same single amino acid substitutions in NS1 as in our in vitro experiments (**Fig 4A**). All mutations resulted in a weaker interaction with MED25 ACID, as compared to WT NS1. The trend of affinity loss was similar to that observed by BLI in vitro. The most significant loss of affinity was observed for the F56A and E110A substitutions, and the least with K112A.

As a control, we carried out the NanoLuc assay with the same NS1-114 constructs and 11S NanoLuc fused to the TRIM25 protein (TRIM25-11S). NS1 was shown to bind to the SPRY domain of the E3-ubiquitin ligase TRIM25, thereby suppressing RIG-I ubiquitination in the cytoplasm, and preventing RIG-I mediated IFN production [12]. Our results show that TRIM25 interacts with NS1 in cells (**Fig 4B**). The Δα3, I54A, F56A, E110A, and M122A NS1 mutations resulted in luminescence comparable to that of WT NS1. Only the NS1 K112A mutation resulted in lower luminescence. These data suggest that neither the NS1 α3 region nor NS1 α/β core domain residues critical for MED25 ACID binding are critical for TRIM25 binding. Neither Alphafold2 nor Alphafold3 generated high-confidence structural models of the NS1–TRIM25 SPRY complex. Since mutated NS1 was still competent for TRIM25 binding, we inferred that the mutations did not inactivate NS1 in cells. Analysis of the lysates by Western blotting using anti-NS1 antibody confirmed comparable protein expression levels for WT and mutated NS1 proteins (**Fig 4C**).

As a last control, we analyzed the cellular localization of NS1 transfected with pCI-neo-NS1 constructs by confocal microscopy, using a bronchial epithelial cell line (BEAS-2B) (**Fig 4D**). BEAS-2B cells were transfected to express WT or mutated NS1 proteins. NS1 localization was determined by confocal imaging after staining with anti-NS1 antibody. All NS1 variants showed cytoplasmic as well as nuclear localization, indicating that none of the mutations affected its nuclear translocation.

Taken together, our data show that the NS1 α/β core domain is necessary for tight binding of NS1 to MED25 ACID in cells, and that the affinity can be significantly reduced by single amino substitutions in this domain, to the same extent as full deletion of α3. This strongly supports the hypothesis of a dual binding site on MED25 ACID.

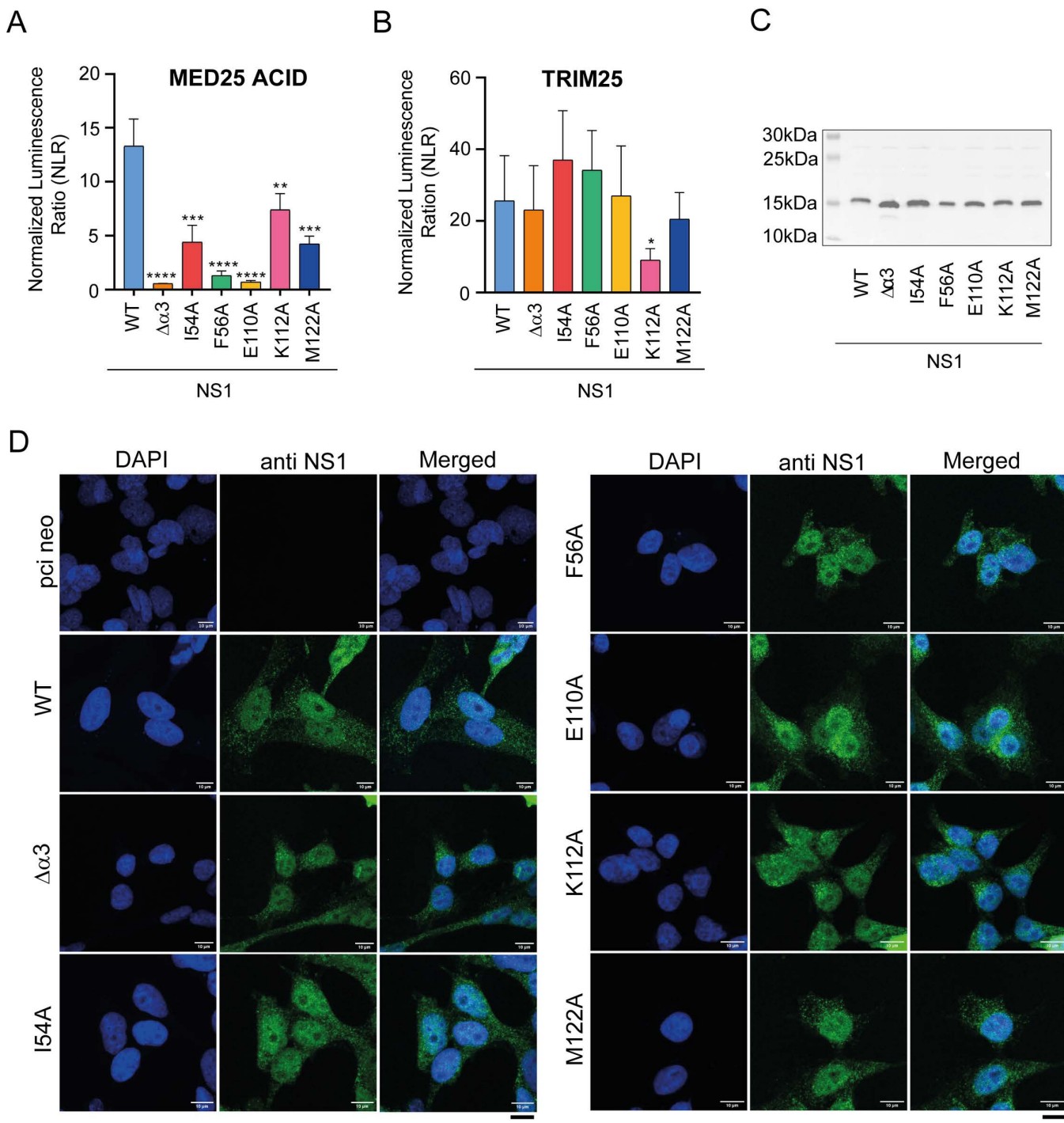

**Fig 4. Interaction in cells between MED25 ACID and WT or mutated NS1 proteins. (A)** Interactions were measured in HEK 293T cells using the NanoLuc assay. NS1 was fused to the NanoLuc 114 subunit (NS1-114) and MED25 ACID to NanoLuc 11S (11S-MED25 ACID). HEK 293T cells were transfected with pairs of constructs, combined as shown on the bar diagram. The normalized luminescence ratio (NLR) is the ratio between actual read and negative controls (each fusion protein with the complementary NanoLuc subunit). Bars represent the mean value of 4 independent biological experiments done in triplicate. Error bars represent the standard deviation. *p<0.05, **p<0.01, ***p<0.001 (unpaired two-tailed t-test). **(B)** Interaction between the TRIM25 protein and NS1, WT or mutated, measured with the NanoLuc assay, using NS1-114 and 11S-TRIM25 constructs. **(C)** HEK 293T cells were transfected with pCI-neo plasmids encoding WT or mutated NS1 constructs fused to 114 NanoLuc. Cell lysates were subjected to Western Blot analysis using an anti-NS1 antibody. **(D)** BEAS-2B cells were transfected with plasmids encoding WT or mutated NS1 constructs fused to 114 NanoLuc. Cells

were fixed, immunostained with anti-NS1 antibody followed by Alexa Fluor secondary antibody (green), and analyzed by confocal fluorescence micros-copy. Nuclear staining was done with DAPI dye (blue). Scale bar = 10 μm.

## Viral replication of recombinant rRSV-mCherry with NS1 mutations

NS1 α3 mutations, i.e., full α3 deletion as well as the Y125A and L132A/L133A substitutions, were previously reported to reduce the replication rate of rRSV in IFN competent lung A549 epithelial cells [38]. We previously showed that the same mutations reduced the interaction with MED25 ACID [26]. We thus wondered if NS1 α/β core domain mutations that disrupt MED25 ACID binding could also attenuate replication. We generated recombinant rRSV-mCherry encoding either WT NS1, NS1 with single amino acid substitutions or NS1 Δα3. Each rRSV-mCherry was recovered in BSR-T7 cells and amplified in Vero cells, which are two IFN-incompetent cell lines. To detect off-target mutations in the virus genomes, we performed next-generation sequencing on all rRSV-mCherry viruses. Amino acid sequences of all RSV proteins were compared between WT rRSV and NS1 mutants. All NS1 mutations were confirmed (S6 Fig). Random mutations were only detected for the M protein of NS1Δα3 rRSV, with three missense mutations (I49T, I62T, and L70P) (S6 Fig), and in the L proteins. All mutants had the L Q177H mutation. The L Y2135H mutation was additionally detected in the NS1 F56A rRSV mutant. These two L protein mutations are normal variants in wildly circulation RSV viruses.

For viral replication experiments, bronchial epithelial BEAS-2B cells were infected with rRSV-mCherry with a multiplic-ity of infection (MOI) of 0.1. BEAS cells were reported to produce an antiviral response associated with IFN-I and IFN-III expression upon RSV infection [48]. Viral replication was visualized with a fluorescence microscope at 6 h, 24 h, 32 h, 48 h, and 56 h post-infection (pi) (Fig 5A). At 32 h pi, mCherry fluorescence was lower for all rRSV NS1 mutants as compared to WT rRSV-mCherry, except for the NS1 I54A mutant, indicating that viral replication was attenuated with respect to WT (Fig 5A). The most attenuated mutant viruses were NS1 Δα3 and E110A rRSV-mCherry. At 56 h pi, all mutated NS1 viruses displayed lower replication than the WT virus. The attenuation of NS1 I54A, F56A, and E110A rRSV-mCherry was com-parable to that of NS1 Δα3 rRSV. NS1 K112A and M122A rRSV-mCherry viruses were the less attenuated. To exclude a default of NS1 protein expression, we analyzed lysates of BEAS-2B cells infected with WT rRSV-mCherry or NS1 mutants by Western blotting, using an anti-NS1 antibody. We found that NS1 protein levels were comparable for all viruses (Fig 5C).

As a control, we measured viral replication in Vero cells, which are not IFN competent. We used WT and three of the most attenuated rRSV-mCherry mutants in BEAS-2B cells, i.e., NS1 I54A, E110A, and Δα3 rRSV-mCherry. Like WT rRSV-mCherry, the three NS1 rRSV mutants induced syncytia formation in Vero cells at 56 h pi (Fig 5B). This is a known cytopathic feature in RSV infected cell lines [49]. Replication of mutant rRSV-mCherry was slightly attenuated compared to WT in Vero cells, in particular at later pi times (Fig 5B). This is in agreement with previous reports for NS1 Δα3 rRSV-mCherry [38]. The attenuation was less significant than in BEAS-2B cells, suggesting that the NS1 rRSV mutants replicate at a comparable level to WT rRSV in cells that do not produce an antiviral response. This points to a correlation between rRSV replication and IFN response for these NS1 mutants. However, the partial attenuation in Vero cells of rRSV-mCherry with the NS1 I54A mutation, which disrupted the NS1–MED25 ACID complex less efficiently than E110A (Fig 4A), indi-cates that alternative antiviral responses may come into play, which are not associated to MED25.

To confirm viral replication data obtained from mCherry fluorescence measurements, we used a viral plaque assay as a complementary method to quantify the virus titer. We infected BEAS-2B cells with either WT or NS1-mutated rRSV-mCherry at an MOI of 0.1. Cell-associated virus samples were collected at 6 h, 24 h, 48 h, and 72 h pi, and viral titers were determined using a plaque assay on Vero cells. Replication of rRSV-mCherry became detectable at 24 h pi, and only minimal differences were observed between WT rRSV and NS1 mutants at this timepoint. At 48 h pi the difference in replication became more pronounced. Consistently with the results obtained by mCherry fluorescence measurements, all NS1 mutants exhibited attenuation compared to WT rRSV-mCherry (Fig 5D). At 72 h pi, the replication rate of the rRSV

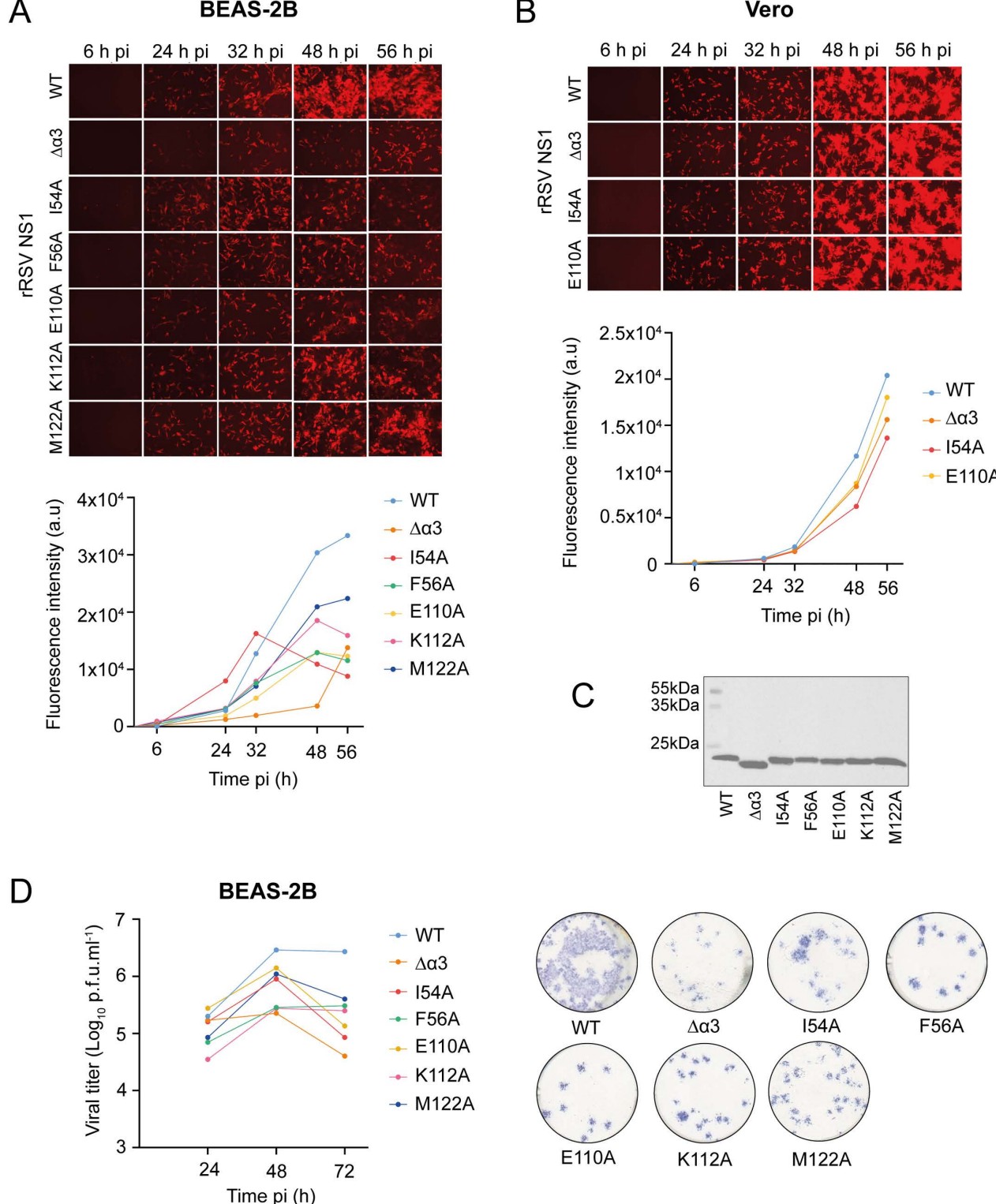

**Fig 5. Viral replication of recombinant rRSV-mCherry coding for NS1 mutated at the MED25 ACID binding interface. (A)** BEAS-2B or **(B)** Vero cells were infected with rRSV-mCherry containing WT or mutated NS1 with a multiplicity of infection (MOI) of 0.1 and visualized by fluorescence microscopy from 6 h to 56 h post-infection (pi). Growth curves were obtained by measuring the fluorescence intensity at the indicated time points pi.

**(C)** Western Blot analysis of lysates of BEAS-2B cells infected with WT rRSV-mCherry or NS1 mutants at an MOI of 3, using an anti-NS1 antibody. **(D)** BEAS-2B cells were infected with WT rRSV-mCherry or NS1 mutants at an MOI of 0.1. Viruses were collected at 6 h, 24 h, 48 h, and 72 h pi, and the viral load was titrated in Vero cells using a plaque assay. Viral spots were revealed with an anti-RSV antibody followed by a secondary antibody coupled with HRP and HRP substrate reagent. One spot represents one virus at 72 h pi. Representation of 1 experiment from 3 independent biological experiments done in duplicate.

NS1 mutants was lower by approximately 1.5 orders of magnitude in BEAS-2B cells as compared to WT (**Fig 5D**). The NS1 Δα3, I54A, and E110A mutants were the most attenuated. Taken together, our data suggest that similarly to the rRSV-mCherry NS1 Δα3 mutant, rRSV-mCherry NS1 α/β core mutants result in attenuated replication in BEAS-2B cells, and that the NS1 mutations that most effectively disrupt the interaction with MED25 result in lower replication rates.

## Impact of NS1 mutations disrupting the MED25 ACID interaction on IFN signaling

It was previously reported that NS1 α3 mutations induced differential gene expression in A549 cells [38]. To examine the potential role of the NS1 α/β core domain versus NS1 α3 in IFN response signaling pathways, we measured expression levels of IFN-I and IFN-III and of selected ISGs in BEAS-2B cells. BEAS-2B cells were infected at an MOI of 3 with WT rRSV-mCherry or with NS1 mutants. Expression was determined by RT-qPCR at 10 h and 24 h pi for IFN-α and IFN-β, and at 24 h pi for IFN-λ (**Fig 6A**). At 10 h pi, relative expression of IFN-α was low as compared to mock infection, and no difference was seen between WT rRSV-mCherry and NS1 mutants. IFN-β levels were still low, but higher than IFN-α levels, upon infection with rRSV-mCherry. No significant difference was observed between the WT and NS1 mutants. At 24 h pi, IFN-β and IFN-λ expression was significantly increased in cells infected with NS1 Δα3 rRSV-mCherry, while it was back to lower levels for the WT and all other mutants. Results obtained for mRNA at 24 h pi were confirmed by measuring IFN-β and IFN-λ protein levels. Only NS1 Δα3 rRSV-mCherry induced higher IFN-β and IFN-λ concentrations (**Fig 6B**).

We next selected ISGs reported to be involved in the cellular response against RSV infection. ISG15, OAS1, IFIT1, and IFIT3 genes were shown to be upregulated upon RSV infection in BEAS-2B cells [48]. IFIT1, IFIT2, IFIT3, IFITM3, and OAS1 genes were shown to be differentially expressed in A549 cells infected with WT RSV and NS1 α3 mutants [38]. The Mx2 gene was more recently reported to be differentially expressed in bronchial cells infected with WT RSV and the NS1 Y125A RSV mutant [45]. BEAS-2B cells were infected at an MOI of 3 with WT rRSV-mCherry or NS1 mutants. Expression of ISGs was determined by RT-qPCR at 10 h pi. WT rRSV-mCherry induced upregulation of the seven ISGs compared to mock infection (**Fig 6C**). Deletion of the NS1 α3 helix had the highest impact, and significantly increased the mRNAs levels of five ISGs, i.e., ISG15, OAS1A, Mx2, IFIT1, and IFIT2, compared to WT rRSV (**Fig 6C**). The M122A mutation at the N-terminal side of the NS1 α3 helix upregulated Mx2. The two most attenuated rRSV-mCherry viruses with NS1 α/β core domain substitutions, i.e., the NS1 E110A and I54A mutants (**Fig 5D**), appeared to be less efficient in controlling antiviral ISG responses than NS1 Δα3 rRSV-mCherry. The NS1 E110A mutation, which strongly impaired MED25 ACID binding, resulted in increased expression of ISG15 mRNA in infected cells. The I54A mutation, which less efficiently disrupted the interaction with MED25 ACID, but attenuated rRSV-mCherry, led to upregulation of three ISGs, ISG15, OAS1A, and IFIT1, compared to WT rRSV. The NS1 F56A and K112A mutations triggered a comparable response to that of WT rRSV-mCherry.

To confirm the RT-qPCR results, we performed Western blotting of two ISGs, IFIT1 and IFITM3, in lysates from BEAS-2B cells infected with WT rRSV-mCherry and the three NS1 Δα3, I54A and E110A mutants (**S7 Fig**). IFITM3 protein levels were comparable for the four viruses, in agreement with the RT-qPCR data. The IFIT1 protein level was increased for the NS1 Δα3 mutant, as compared to WT, consistently with mRNA measurements. The I54A mutant resulted in a lower IFIT1 concentration, whereas the E110A mutant led to a higher concentration, showing that the IFIT1 mRNA levels do not reflect the protein levels for this ISG. Altogether, these data suggest that NS1 α3 plays a role in IFN production and signaling. However, this may not be mediated by MED25, since the NS1 I54A and E110A α/β core domain mutants, which

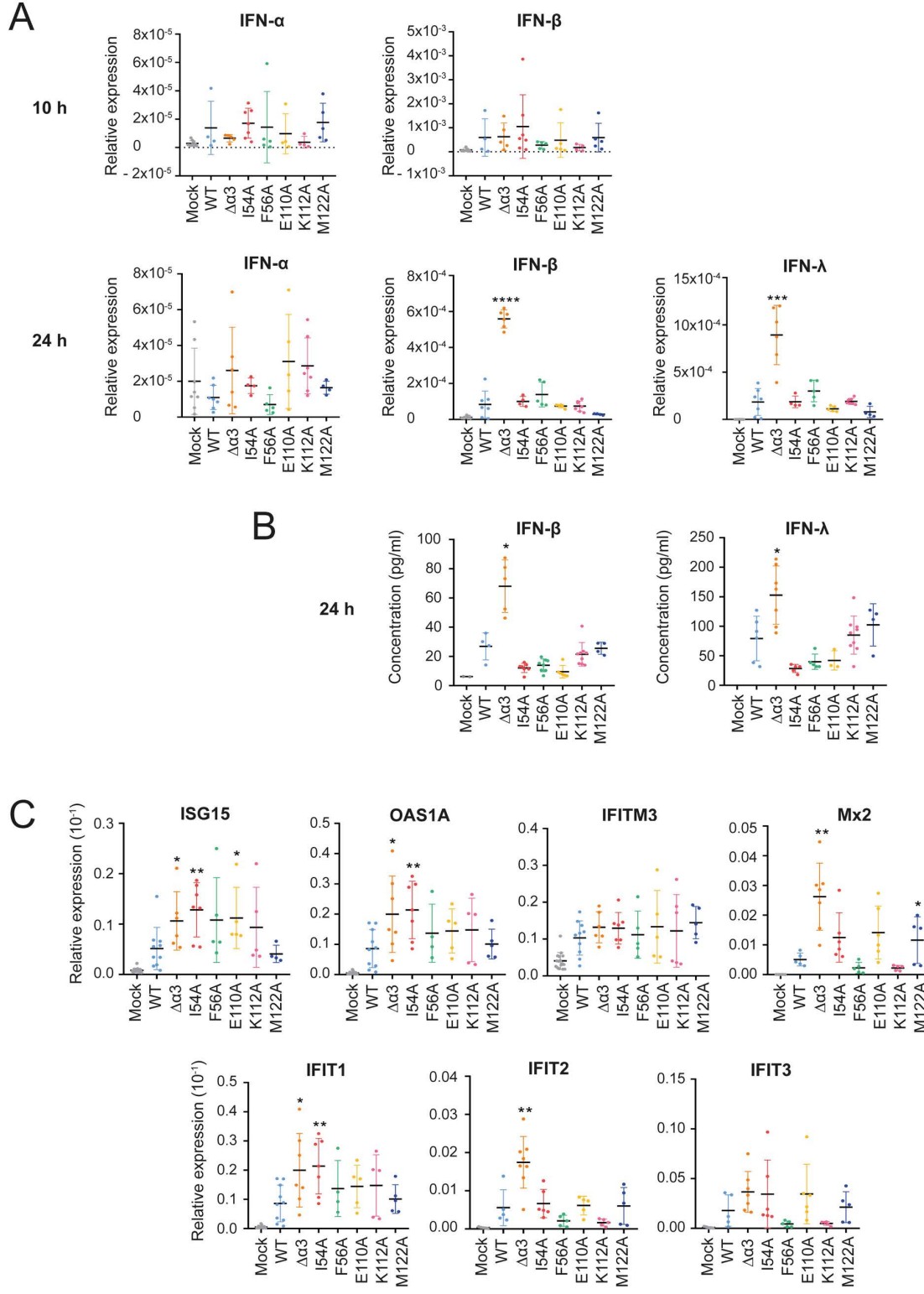

**Fig 6. Expression levels of interferon-stimulated genes (ISGs) in BEAS-2B cells infected with mock or rRSV-mCherry coding for NS1 variants.** BEAS-2B cells were infected with rRSV-mCherry, coding for WT or mutated NS1, with an MOI of 3. **(A)** mRNA expression of type I IFN (α and β) and

type III IFN (λ) was quantified by RT-qPCR at 10 h and 24 h pi. **(B)** The concentration of IFN-β and IFN-λ proteins in the supernatant was measured by Luminex at 24 h pi. **(C)** Expression levels of ISG mRNA were determined by RT-qPCR at 10 h pi under the same conditions as in **(A)**. For the RT-qPCR, the fold change was calculated with the ΔCt method. Data are from at least 4 independent experiments. Mean ± standard error of the mean are represented. *p < 0.05, **p < 0.01 (Mann-Whitney test).

also disrupt the MED25 interaction in cells, display a different behavior. They do not stimulate IFN production, but appear to induce higher levels of specific ISGs.

## Comparison of WT and mutant NS1 rRSV properties in MED25-depleted cells

To specifically examine the role of MED25 during RSV infection for NS1-mediated antiviral responses, we measured rRSV replication in MED25-knockdown A549 cells. A549 cells produce innate antiviral responses upon RSV infection [48]. This cell line has been used previously to study the effect of MED25 knockout on WT RSV replication [27]. Transfection efficiency is higher in A549 cells than in BEAS-2B cells. We therefore transfected A549 cells with siRNA against MED25, and subsequently infected the transfected cells with WT rRSV-mCherry or NS1 mutants with an MOI of 0.5. A negative control siRNA was used for comparison. mCherry fluorescence was measured at 48 h pi to assess rRSV-mCherry replication. Western blot analysis of MED25 expression in cells transfected with siRNA against MED25 versus control siRNA confirmed the efficient depletion of MED25 (**Fig 7A**). No significant toxicity or cell death, as measured by quantifying ATP, was detected, when comparing control and MED25-depleted cells (**Fig 7B**). In negative control A549 cells, in which MED25 was present, viral replication was significantly attenuated for the NS1 Δα3 mutant compared to WT rRSV-mCherry (**Fig 7C**). This is consistent with previously reported data obtained with recombinant RSV A2 line 19F [38]. The rRSV-mCherry mutants with NS1 I54A and E110A mutations were also significantly attenuated. This mirrors the decrease of viral fitness for the three mutants compared to WT rRSV-mCherry observed in BEAS2B cells (**Fig 5A** and **5D**). In MED25-knockdown A549 cells, rRSV-mCherry replication was decreased for WT, as compared to the negative control sample. This was surprising, since Van Royen et al. had observed increased RSV replication in MED25-knockout A549 cells, when using the RSV B1 strain [27]. The three rRSV NS1 mutants were also more attenuated, compared to the negative control cells. However, in contrast to the control experiment, the replication difference between WT and mutant rRSV was less significant, in particular for the I54A and E110A mutants.

To test the influence of our experimental setting, we measured replication on a second recombinant virus derived from the RSV A2 strain line 19F, containing an mKate2 reporter [50]. rRSV-mKate2 replicated to similar titers in MED25-depleted A549 cells at 48 h pi and in control cells (S8A Fig), showing that the effect of MED25 could be strain dependent. We also investigated earlier timepoints for rRSV-mCherry, at 10 h and 24 h pi (S8B Fig). At 10 h pi, replication was fully restrained in negative control cells, whereas low titers were measured for the four viruses in MED25-depleted cells. At 24 h pi, WT rRSV-mCherry replicated to similar titers in control and in MED25-depleted A549 cells. The rRSV-mCherry NS1 E110A mutant was significantly attenuated in control cells as compared to WT rRSV-mCherry. The two other mutants replicated to similar titers as WT rRSV-mCherry in control cells. In MED25-knockdown A549 cells, the two rRSV-mCherry NS1 E110A and Δα3 mutants were significantly attenuated with respect to WT. In contrast to the 48 h pi timepoint, the attenuation of the NS1 E110A rRSV-mCherry mutant was more significant in MED25-knockdown A549 cells than in control cells. Taken together these results show that there is a MED25-dependent effect on rRSV-mCherry replication, which is modulated by mutations disrupting the NS1–MED25 ACID complex, in a time-dependent manner.

To rationalize the decrease in rRSV-mCherry titer in MED25-knockdown cells at 48 h pi, we measured the expression levels of four antiviral ISGs. A549 cells were transfected with MED25 or control siRNA, and infected at an MOI of 2 with WT rRSV-mCherry or NS1 mutants. Expression of ISGs was determined by RT-qPCR at 48 h pi. For IFITM3 and OAS1A, only minimal differences of expression levels were observed in A549 cells infected with WT or rRSV-mCherry mutants,

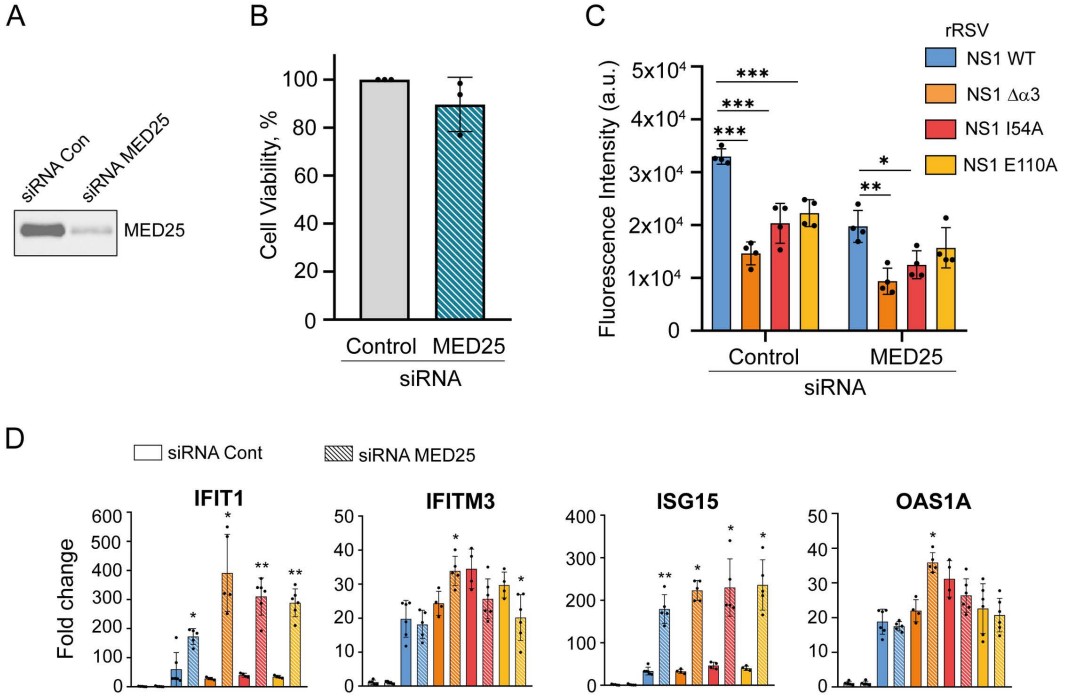

**Fig 7. Impact of MED25 knockdown on rRSV replication and ISG expression in A549 cells.** A549 cells were transfected with 10 nM of control siRNA or siRNA against MED25, followed by infection 24 h later with WT or NS1 mutant rRSV-mCherry viruses at an MOI of 0.5. **(A)** Western blot analysis of MED25 depletion was performed 48 h pi in A549 cells transfected with control versus MED25 siRNA. **(B)** Cellular viability was quantified at 48 h pi in A549 control cells compared to A549 MED25-knockdown cells with a bioluminescence assay. The figure represents 3 independent experiments. **(C)** RSV replication was quantified at 48 h pi by measurement of mCherry fluorescence intensity. Data show means and standard errors of 4 independent experiments, p < 0.05 *, p < 0.005 **, p < 0.0005 *** (t-test). **(D)** Expression levels of ISG mRNA, determined by RT-qPCR 48 h pi, in MED25-knockdown A549 cells infected with WT or NS1 mutant rRSV-mCherry at an MOI 2. The fold change was calculated using the ΔCT method. Data are from at least 4 independent experiments. Mean ± standard error of the mean are represented. *p < 0.05, **p < 0.01. *p < 0.05, **p < 0.01 (Mann-Whitney test). The statistical test was used to compare the control siRNA condition with the MED25 siRNA condition for each rRSV-mCherry virus.

when MED25 was knocked down as compared to the control experiment (**Fig 7D**). For the two other ISGs, IFIT1 and ISG15, a drastic increase of mRNA was observed in MED25-knockdown cells, compared to negative control, for WT rRSV-mCherry and the three NS1 mutants. This increased antiviral response could partly explain the lower replication of WT and mutant rRSV-mCherry in MED25-knockdown cells at 48 h pi (**Fig 7C**). However, we did not observe significant differences in ISG expression levels upon infection by rRSV-mCherry NS1 mutants compared to WT. Overall, our results suggest a correlation between NS1 binding to MED25, RSV replication and restriction of antiviral responses in A549 cells. They also suggest that MED25 is required for RSV in order to control cellular transcription of ISGs. However, this effect appears to be time-dependent and RSV strain-specific.

## Discussion

### RSV NS1 protein achieves tight binding to MED25 ACID *via* a dual binding site

Mammalian MED25 was shown to be targeted by the cellular transcription activators ERM [33,51], ATF6α [52], p53 [37], and ETV4 [34], as well as Herpes simplex virus (HSV) VP16 [29–32], Varicella-zoster Virus (VZV) IE62 [53], and Kaposi's sarcoma-associated herpesvirus (KSHV) lana-1 viral proteins [54]. NMR was used to map the binding regions for the corresponding TADs to the H1 or H2 interfaces of MED25 ACID. When transcription activators possess two TADs, like p53

and VP16, each TAD binds specifically to one of these interfaces. In the case of VP16, the two TADs activate transcription independently and bind cooperatively to the H1 and H2 faces of MED25 ACID [29,30]. MED25-specific TADs are variable in composition, but they all possess short motifs that are unstructured in isolation and are assumed to adopt an α-helical conformation in complex with MED25 ACID. To date there is no experimental high-resolution structure of a MED25 ACID−TAD complex. Even structure prediction with deep learning methods remains challenging. Binding of the NS1 α3 peptide, which reminds of a TAD, could not be resolved by AlphaFold2, which proposed models with NS1 α3 binding either to MED25 ACID H1 or to H2 [28]. It was therefore surprising to obtain high confidence structure models for the NS1–MED25 ACID complex by AlphaFold. In the context of full-length NS1, NS1 α3 acts like an H1-binding TAD despite its preference for the H2 interface as a peptide. The NS1 α/β core domain, which shares no structural similarity with a TAD, also binds to MED25 ACID. Experimental evidence by NMR supports the hypothesis that it binds to the MED25 ACID H2 interface. Both subdomains contribute to an extensive interaction surface. These findings were recently confirmed by X-ray crystallography [45].

The difference between the affinities of single NS1 subdomains versus full-length NS1 shows that binding is cooperative, as observed previously with VP16 TAD1 and TAD2 [29,30]. Strikingly, NMR perturbations of $^{15}$N-ME25 ACID by NS1 were observed in the two antiparallel α1 and α3 helices of MED25 ACID (**Fig 2E** and **2F**). These are ideally positioned to mediate coupling between H2 and H1, likely via an allosteric communication, as proposed earlier [36]. Interestingly, binding kinetic measurements by BLI showed that the complex between MED25 ACID and NS1Δα3, i.e., the isolated NS1 α/β core domain, does not adopt a single binding mode. This was also observed for the single amino acid mutants of full-length NS1. A first binding mode (mode 1) displayed faster association and faster dissociation, concomitantly with lower affinity for amino acid substitutions (**Table 2**). This would be in line with the formation of an encounter complex, with suboptimal binding geometry. The first binding mode was the most populated for the NS1 mutants that most efficiently disrupted the complex. A second binding mode (mode 2) displayed slower association, but also slower dissociation, and higher affinity. This could be explained by structural rearrangements in the encounter complex. However, it cannot be ruled out that association occurs via conformational selection. For the NS1 Δα3 mutant, mode 1 displayed higher affinity. It can be thus assumed that the α3 subdomain is key to proceed to mode 2. Overall, our results suggest that mutations that disrupt the NS1–MED25 ACID complex kinetically trap the encounter complex.

Kds for MED25 ACID complexes with a single TAD were reported in the µM range (0.5 µM for ATF6α TAD [36], 0.6 µM for ERM [33,36], 1.6 µM for VP16 TAD1 [30], 8.1 µM for p53 TAD2 [37]), while Kds for tandem TAD1-TAD2 TADs were in the 50 nM-1 µM range (50 nM for VP16 TAD [30,36], 0.8 µM for p53 TAD [37]). NS1 binds to MED25 ACID in the high affinity range for TADs (Kd 16 nM by BLI, and 44 nM by ITC). According to the AlphaFold structure prediction, NS1 fully occludes the TAD-binding sites. This strongly suggests that NS1 can efficiently compete with cellular TADs for binding.

## Impact of RSV NS1 α3 and α/β subdomains on rRSV replication

Based on the AlphaFold complex model, we designed five alanine substitutions in NS1 at the MED25 ACID interface. NS1 E110A and F56A substitutions were the most efficient to disrupt the interaction in vitro (Tables 2 and 3). In the AlphaFold model, NS1 E110 is close to MED25 ACID K518. However, the position of sidechains is not optimal for an electrostatic interaction, as can also be inferred from the X-ray crystal structure (PDB 9ccv), which rather suggests that a network of hydrogen bonds links the two proteins in this region. The proximal NS1 K112 residue does not appear to be crucial for the complex, contrary to what was suggested based on the X-ray structure [45]. Conversely, F56 is close to MED25 ACID M523, and we showed previously that the M523E mutation disrupted the complex in vitro [26]. In contrast to in vitro experiments with recombinant proteins, all NS1 mutants displayed significantly lower affinity in cellula (**Fig 4A**). In cell interaction experiments confirmed the potency of the two single amino acid mutations in the NS1 α/β domain, E110A and F56A, to disrupt the interaction to the same extent as the deletion of the α3 helix. In cellula, the three NS1 mutants with the lowest MED25 ACID affinity still bound to another NS1 protein partner, TRIM25 (**Fig 4B**). This confirms that these

mutations specifically affected the interaction with MED25, and that the cytosolic function of NS1 remained intact for the rRSV mutants. Moreover, in IFN-deficient Vero cells, recombinant rRSV with NS1 Δα3 and E110A mutations displayed similar syncytia formation, and replication rates were comparable to those of the WT virus (Fig 5B). This indicates that these NS1 mutations did not affect any structural aspects of the virus or the capacity for viral replication in the absence of an innate immune response.

In contrast, in BEAS-2B cells, which produce an antiviral response associated with the expression of IFN-I and IFN-III [48], all mutant rRSV viruses showed less syncytia formation. They also displayed an approximately 1–2 orders of magnitude decrease in replication compared to WT rRSV-mCherry at time points later than 32–48 h pi (Fig 5A and 5D). All six mutated NS1 proteins showed intact nuclear localization, similarly to WT NS1 (Fig 4), indicating that the defects were not due to impaired NS1 cellular localization. A rough correlation can be drawn between the loss of MED25 ACID binding affinity and the loss of viral fitness, since the NS1 K112A and M122A mutations, which retained the highest affinities, also led to less attenuated rRSV-mCherry, while NS1 Δα3 and E110A rRSV-mCherry mutants were the most attenuated (Fig 5A and 5D). However, since NS1 is a small multifunctional and multipartner protein [10,11,19], it cannot be excluded that amino acid changes in the MED25 ACID binding interface of NS1 also affect interactions with other partners, relieving inhibition of alternative pathways of the innate immune response. In particular, rRSV-mCherry with the NS1 I54A mutation, which is less critical for the interaction with MED25 ACID, is attenuated, and its infection dynamics differ from that of the other mutants at 48–56 h pi (Fig 5A). This would also be in line with the NS1 I54A rRSV mutant being more attenuated in IFN-incompetent Vero cells, compared to Δα3 and E110A mutants (Fig 5B).

Of note, sequencing of NS1 Δα3 rRSV-mCherry revealed three additional mutations in the matrix (M) protein (S6 Fig). The M protein, which is involved in virion assembly and budding, is a structural paralog of NS1 [38]. The lower viral titers measured for NS1 Δα3 rRSV-mCherry at 48 h pi, compared to the other mutants, are most likely a direct consequence of the NS1 mutation that affects MED25 binding. But, we cannot rule out that the detected M mutations also contribute to reduced viral fitness.

## Impact of the RSV NS1–MED25 interaction on the innate immune response triggered by RSV

To investigate the origin of the reduced viral replication of NS1 mutants, we checked IFN induction by rRSV-mCherry infection. We focused on type I and III IFNs that play a crucial role during early RSV infection, mostly IFN-β and IFN-λ(48). IFN-I is known for its anti-proliferative and immunomodulatory effects, while IFN-III induces supplementary protection at the surface of epithelial cells [55]. Our results demonstrate that only the NS1 Δα3 rRSV mutant significantly upregulates IFN-I and IFN-III, as evidenced by mRNA expression, corroborated by protein concentration measurements (Fig 6A and 6B). We conclude that the NS1 α3 helix is crucial for IFN production. However, since other NS1 mutations that disrupt the NS1–MED25 ACID complex do not have the same effect, the deletion of α3 likely affects other pathways for IFN induction, not mediated by MED25. In particular, NS1 interacts with several cytoplasmic host factors involved in IFN induction signaling [10,11]. NS1 directly targets the transcription factor IRF3, leading to inhibition of IFN production [56,57]. NS1 also interferes with RIG-I activation by targeting the E3 ubiquitin ligase TRIM25 [12], but not likely via NS1 α3 as shown here (Fig 4B).

Next, we investigated IFN responses, and focused on ISGs previously shown to be upregulated in RSV-infected BEAS-2B cells [48] or differentially expressed in A549 cells [38,45]. Proteins of the IFIT family, such as IFIT1, IFIT2, and IFIT3 act as a sensors of viral RNA, thereby inhibiting viral replication [58,59]. IFITM3 restricts cellular entry for several viruses [60]. ISG15 is a ubiquitin-like protein that leads to ISGylation of several target proteins involved in the activation of innate immune response [61]. OAS1A is an oligoadenylate synthetase that inhibits viral replication via activation of RNase L [62]. IFIT2 and IFIT3 were shown to be in the proximity of promoters targeted by the NS1–MED25 complex and were among the most differentially expressed genes during RSV infection [21]. IFIT1 and IFITM3 proteins were shown to be targets of RSV NS1, with significantly reduced levels in HEK 293 cells in the presence of RSV NS1 [63]. OAS1 and IFITM3

were shown to be upregulated in RSV-infected A549 cells [38,64]. Mx2 was recently identified as a differentially expressed gene in RSV infected cells versus mock, regulated by the ATF3 transcription factor and modulated by the NS1–MED25 interaction [45]. Our results show that gene expression for the six out of seven ISGs is upregulated upon infection with rRSV-mCherry, as compared to mock-infected BEAS-2B cells (Fig 6C). IFITM3 did not appear to be significantly affected. Expression of five ISGs, in particular Mx2, was further upregulated by the NS1 Δα3 deletion. Among the NS1 α/β core mutations, only I54A and E110A led to upregulation of 1–3 ISGs. Overall, the restriction of viral replication in BEAS-2B cells, as measured by viral titer and syncytia formation (Fig 5A and 5D), appears to be correlated to IFN signaling rather than to IFN production.

We observed that rRSV-mCherry replicated less in MED25-knockdown A549 cells at 48 h pi (Fig 7B). This stands in contrast to previously published data that showed enhanced RSV replication in MED25-knockout cells, suggesting an antiviral effect for MED25 [27]. We verified that depletion of MED25 by siRNA had no toxic effect on cells, and concluded that the decrease in viral replication was not due to a negative effect on cell function. siRNA was used previously to knock down MED25 in other cell lines like U2OS cells [51,52], based on the observation that MED25 is a labile Mediator component [65]. The difference between the two studies may be due to the completeness of MED25 depletion. We used a knockdown approach, whereas Van Royen et al. used CRISPR-Cas9 to knock out MED25. This might affect the Mediator complex function, although MED25 was identified as a non-essential Mediator subunit [66]. Data at earlier time points showed that the relative viral titer was time dependent for rRSV-mCherry from RSV Long strain (S8 Fig). To compare with another recombinant virus, we used rRSV-mKate2 from the RSV A2 strain. It replicated to similar titers in MED25-depleted A549 cells and control cells. This is compatible with the previous report: replication of the laboratory RSV B1 strain was enhanced in MED25-knockout cells, but replication titers of RSV A2 strain titers were close in MED25-knockout and WT A549 cells [27]. Altogether, this suggests that the effect of MED25 depends on RSV strains.

Viral replication was significantly attenuated for the three NS1 Δα3, I54A, and E110A rRSV-mCherry mutants, compared to WT rRSV-mCherry, in BEAS-2B cells (Fig 5) as well as in A549 cells (Fig 7). In contrast, in MED25-knockdown A549 cells, the replication difference between WT and the mutant rRSV-mCherry viruses was less significant at a later timepoint (Fig 7C), suggesting that in the absence of MED25, the antiviral response was comparable for these viruses. Since NS1 Δα3 and E110A mutations disrupted the interaction with MED25, these results provide first evidence for MED25 hijacking by RSV to reduce the cellular immune response upon RSV infection.

## Hypotheses for a competition mechanism between RSV NS1 and cellular TADs in targeting MED25

Hijacking of the MED25 Mediator subunit has been reported for other viruses, with very different mechanisms. HSV VP16 [67,68] and VZV IE62 [53,69] are genuine immediate-early transcription activators, containing DNA-binding domains and TADs that bind to MED25. KSHV Lana-1 was shown to act differently. It induces the serum response element by acting as an adaptor protein connecting the serum response factor and the Mediator via MED25 in a ternary complex [54]. It was proposed more generally that viral transcription regulators could directly bind to host regulators rather than to nucleic acids [40]. Very recently, the hypothesis of a ternary complex involving NS1, MED25 and the stress induced ATF3 transcription factor was raised [45]. In a comprehensive chromatin binding study, ATF3 was found to be enriched at genes whose expression was significantly altered by RSV infection and by the NS1 Y125A mutation, in particular immune response genes. Moreover, ATF3 was found proximal to the NS1–MED25 complex [45]. In this study, ATF3 transcription activation was reduced in cells overexpressing NS1. Transfection of cells with a plasmid encoding NS1 with the triple mutation E110A/L132A/L133A, which fully impaired MED25 ACID binding, partially restored transcriptional activity in an ATF3-dependent manner. The presence of NS1 decreased the affinity of DNA-bound ATF3 for MED25 ACID. Due to the extremely low affinity of the ternary complex, an alternative explanation for the loss of function of ATF3 due to NS1 would be the inhibition of ATF3 by competitive binding of NS1 to MED25. In addition, ATF3 was identified by comparing WT with an RSV mutant containing the NS1 α3 helix Y125A mutation.

NS1 Y125A retains high affinity for MED25 ACID, as shown here, but may disrupt interactions with other NS1 partners, which have not been structurally characterized yet.

We previously showed that NS1 competes in vitro with the TAD of human activating transcription factor 6 (ATF6α) for MED25 binding [26]. ATF6α is a sensor of misfolded protein. In response to endoplasmic reticulum (ER) stress, it is transported into the nucleus to activate transcription of ER stress response genes via a direct interaction with the MED25 subunit [35,52]. The rationale for using this specific transcription factor was that RSV was shown to induce a non-canonical ER stress response via activation of AFT6 pathways in A549 cells [70]. Moreover, the H2 interface of MED25 ACID was reported to be the interaction site of the ATF6α TAD [36]. Recently, competition for MED25 ACID binding was demonstrated between ATF6α and an ad hoc lipidated H2-binding peptide with TAD-like amino acid composition [71]. This peptide, with an estimated Kd of 4 μM, was able to partially inhibit MED25-dependent gene transcription [71]. Hence, tight binding of NS1 to MED25 ACID in the 20–50 nM range as well as occlusion of the H2 interface by the NS1 α/β core domain rationalize NS1 competition with ATF6α binding and signaling. The high affinity of NS1 for MED25 ACID and the dual binding domain on MED25 ACID suggest that NS1 would efficiently compete with cellular H1-binding TADs as well as with H2-binding TADs.

Transcription inhibition due to competitive binding has also been demonstrated for p53-dependent transcription, with a stapled peptide mimicking the p53 tandem TAD, which blocked p53-Mediator binding and suppressed the p53 response [72]. More generally, the high affinity of the NS1–MED25 ACID complex emphasizes the relevance of MED25 as a cellular target of NS1 for host transcription regulation by RSV. The downstream effect of the sequestration of MED25 by NS1 is still unclear. Several cellular transcription factors were reported to target MED25: RAR, HNF4α, ERM, SOX9, and ATF6α [22]. Since RSV NS1 acts on multiple pathways that more directly intervene in IFN-I and IFN-III production and signaling, the interaction with MED25 is not strictly necessary to regulate antiviral protein levels. For example, HNF4α controls multiple metabolic pathways, in particular lipid homeostasis [73]. The interaction with MED25 may therefore participate in modeling a general antiviral and inflammatory state of the cell upon RSV infection. Additional studies are required to identify the specific downstream pathways affected by the NS1–MED25 interaction.

## Materials and methods

### Plasmids

The MED25 ACID (NCBI Gene ID 81857, MED25 aa 389–543) construct was obtained by introducing start and stop codons at the appropriate site in Addgene #64771 plasmid, as described previously [26]. The sequence of NS1 is a variant from RSV subgroup A Long strain (GenBank AY911262) containing the N102D mutation. For NanoLuc plasmids, custom synthesized pCI-neo NanoLuc 114 and 11S vectors (Promega) were used to clone the codon-optimized RSV NS1, MED25, and TRIM25 (GeneCust) constructs using standard PCR, digestion, and ligation techniques. pCI-neo-NS1 single-site mutations were obtained by using the QuikChange site-directed mutagenesis kit (Stratagene). NS1 Δα3 (aa 1–118) was generated by introducing start and stop codons at the appropriate site in the coding sequence using standard PCR, digestion, and ligation techniques. A pET28 plasmid was used to express recombinant His-tagged MED25 ACID (aa 389–543) protein. pGEX-4T3 plasmids were used to express recombinant WT and mutated GST-tagged NS1 proteins. Mutations were introduced by site-directed mutagenesis using the QuikChange kit (Stratagene). NS1Δα3 (aa 1–116), used for NMR, ITC and BLI measurements, was constructed by introducing a stop codon at amino acid position 117. Primers are listed in S1 Table.

### Cells

Vero (ATCC number CCL-81), HEK 293T (ATCC number CRL-3216), BSRT7/5, and A549 (ATCC number CCL-185) cells were maintained in Dulbecco's modified Eagle's medium (DMEM). BEAS-2B cells (ATCC number CRL-3588) were

maintained in RPMI 1640 medium (Eurobio Scientific), supplemented with 10% fetal bovine serum (FBS-12A, Capricorn Scientific), 1% L-glutamine (200 mM) (Eurobio Scientific), and 1% penicillin-streptomycin (100 U/ml and 100 µg/ml, respectively). The cells were grown at 37°C in the presence of 5% $CO_2$.

## Virus strains and recombinant viruses

Recombinant rRSV-mCherry viruses were rescued by reverse genetics using a pACNR1180 vector [74], constructed from human RSV subgroup A, Long strain (GenBank AY911262) and coding for the whole rRSV-mCherry antigenome, as previously described [75], and amplified in Vero cells. Recombinant monomeric Katushka 2 (mKate2) human A2 line 19F RSV was rescued by reverse genetics, as previously described [50]. The NS1 with single amino acid mutations or with the α3 deletion were generated on pcDNA cassette using the QuikChange II Site-Directed Mutagenesis kit (Stratagene) and subcloned into the pACNR1180-rRSVmCherry vector [74] using the In-Fusion HD Cloning Kit (Takara). Specific primers containing AfeI and KpnI restriction enzyme sites were used for In-Fusion PCR. All primers are listed in S2 Table. WT and NS1 mutant rRSV-mCherry genomes were sequenced by next-generation sequencing, as previously described [76]. RNA from WT or mutant rRSV-mCherry was extracted using the TRI Reagent (Molecular Research Center) and treated with RQ1 RNase-Free Dnase (Promega). One microgram of each viral RNA was reverse-transcribed into cDNA with the Lunascript RT Supermix 5X kit (NEB) and amplified with Q5 High-Fidelity PCR Kit (NEB).

## Virus fluorescence quantification

BEAS-2B cells or Vero cells were seeded at 1 x $10^5$ cells per well in 24-well plates the day before infection. Cells were infected with WT or mutated NS1 rRSV-mCherry at a multiplicity of infection (MOI) of 0.1 in RPMI or DMEM SVF-free medium. At 6 h, 24 h, 32 h, 48 h, and 56 h post-infection, the red fluorescence was measured using an Infinite M200 PRO microplate reader (TECAN, Männedorf, Switzerland) with excitation and emission wavelengths of 580 and 620 nm, respectively. The fluorescence intensity of the infected cells was normalized with the fluorescence of uninfected BEAS-2B or Vero cells (Mock).

## Virus plaque assay

All rRSV-mCherry viruses were titrated on Vero cells by standard plaque assay: cells were infected with serial 10-fold dilutions of viral suspension in DMEM SVF-free medium. The overlay consisted of cellulose Avicel RC581 (FMC Biopolymer) at a final concentration of 1.2% in 2X MEM. After 7 days of infection at 37°C, 5% $CO_2$, cells were fixed and incubated with a monoclonal mouse RSV A/B antibody (Chemicon, MAB858-4) (1:2000) and horseradish peroxidase (HRP)-conjugated goat anti-mouse antibody (Sera Care) (1:2000). Plaques were revealed using the KPL TrueBlue Substrate (Sera Care), and the number of plaque-forming units (p.f.u.) per well was counted visually. The mock control consisted of Vero cell culture supernatant.

## Analysis of RSV NS1 protein sequence

NCBI Virus blast was used to scan sequence conservation in RSV NS1 using the Protein Search mode. The server (https://www.ncbi.nlm.nih.gov/labs/virus/vssi/#/) was accessed on September 19, 2024.

## Structure predictions

AlphaFold2 predictions were run using the python notebook AlphaFold2.ipynb available through the ColabFold [77] interface (https://colab.research.google.com/github/sokrypton/ColabFold/), accessed on June 30, 2023. AlphaFold3 predictions were obtained from the AlphaFold server (https://alphafoldserver.com/), accessed on May 29, 2024. The protein query sequences were MED25 ACID (Uniprot Q71SY5, aa 389–543), full-length NS1 (aa 1–139), and C-terminally truncated

NS1 (aa 1–116). Visualization of the PAE for predicted complex structures was done with the PAE Viewer webserver [78]. Figures were prepared with ChimeraX [43] or with Pymol 3.1.1 [79] software.

## Expression and purification of recombinant proteins

MED25 ACID, containing an N-terminal 6xHis-T7 tag, was expressed from *E. coli* BL21(DE3) bacteria transformed with a pET28-MED25 ACID plasmid. Bacteria were grown from a 20 mL starter culture in 1 L 2YT medium at 37°C to an optical density of 0.6 at 600 nm. Induction was made with 0.1 mM isopropyl-β-D-thiogalactoside (IPTG) for 24 h at 20°C. $^{15}$N-labeled MED25 ACID was produced in minimal M9 medium supplemented with 1 g.L$^{-1}$ $^{15}$NH$_4$Cl (Eurisotop, France). Bacteria were lysed by sonication in 50 mM Na phosphate (NaP), 300 mM NaCl, 10 mM imidazole, pH 8 buffer containing protease inhibitors (complete, Roche), and 1 mg.mL$^{-1}$ lysozyme (Thermo Fisher). Lysates were clarified by ultracentrifugation. His-tagged MED25 ACID (His-MED25 ACID) was captured with 2 ml of Ni-NTA resin (Thermo Fisher). The protein was stepwise eluted with NaP/NaCl buffer containing increasing amounts of imidazole (25, 50, 500 mM). The protein buffer was then exchanged to 20 mM Tris pH 8.0, 150 mM NaCl buffer, using a Hiprep Desalting 26/10 column (GE Healthcare). 1 mM dithiothreitol (DTT), 2.5 mM CaCl$_2$, and 5 units/mL thrombin (Sigma) were added to cleave the tag, and the mixture was incubated at 4°C overnight. Alternatively, purification of His-tagged MED25 ACID was carried out by loading the clarified lysate on HisTrap FF 1 mL or 5 mL columns, and eluting His-tagged MED25 ACID with an imidazole gradient in 50 mM NaP, 25 mM imidazole, 500 mM NaCl pH 8.0 buffer on an Akta System. All samples were further purified by gel filtration on a Superdex S75 Hiload 16/600 column (Cityva) equilibrated with 20 mM NaP pH 6.5, 100 mM NaCl buffer. 5 mM DTT or TCEP was then added, and the protein was concentrated to 200–500 µM using centrifugal filter units with a 10 kDa cut-off (Amicon Ultra, Millipore). The MED25 ACID concentration was determined by measuring the absorption at 280 nm and using the theoretical molar extinction coefficient calculated on the ProtParam server (https://web.expasy.org/protparam/).

WT and mutated NS1 were expressed from *E. coli* BL21(DE3) transformed with pGEX-NS1 plasmids. Cultures were grown from 20 mL starter cultures in 1 L LB or 2YT medium at 37°C to an optical density of 0.6-0.8 at 600 nm, and induced with 0.3 mM IPTG for 24 h at 20°C. After harvesting, cells were first incubated in 20 mM Tris pH 8, 300 mM NaCl, 5% glycerol buffer, supplemented with protease inhibitors (complete, Roche), 1 mg.mL$^{-1}$ lysozyme (Thermo Fisher), 0.2% Triton X100, 10 mM MgSO$_4$, and 2500 U benzonase (Sigma-Aldrich). 1 mM DTT and up to 1.5 M NaCl were added, and the bacteria were sonicated. Lysates were clarified by ultracentrifugation. GST-NS1 was captured with 2 ml Glutathione Sepharose beads (GE Healthcare). The beads were extensively washed with 20 mM Tris pH 8, 300 mM NaCl, 5% glycerol. They were incubated overnight with 20 units of thrombin (Sigma-Aldrich) at 4°C in 20 mM Tris, 150 mM NaCl, 2.5 mM CaCl$_2$, and 5 mM 2-mercaptoethanol. NS1 was eluted and further purified by gel filtration on a Superdex S75 Hiload 16/600 column (Cityva) equilibrated with 20 mM Tris pH 8, 200 mM NaCl, 1 mM DTT buffer. 5 mM TCEP was added to the fractions containing NS1, and the protein was concentrated to 200–500 µM using centrifugal filter units with a 10 kDa cut-off (Amicon Ultra, Millipore). The NS1 concentration was determined from absorption at 280 nm.

## SDS-PAGE and Western blot analysis

Protein samples were separated by electrophoresis on 15% polyacrylamide gels in Tris-glycine buffer. All samples were boiled for 3 min prior to electrophoresis. Proteins were then transferred to a nitrocellulose membrane (Roche Diagnostics). The blots were blocked with 5% non-fat milk in Tris-buffered saline (pH 7.4), and incubated with monoclonal rabbit NS1 (GeneTex, GTX638591) (1:1000), IFITM3 (Ozyme, D8E8G) (1:1000), or IFIT1 (Ozyme, D2X9Z) (1:1000) antibodies or monoclonal mouse α-tubulin (Millipore, T6199) (1:1000) antibody, and horseradish peroxidase (HRP)-conjugated goat anti-rabbit antibody (1 mg/ml, 5450–0010 Seracare) (1:10,000). Western blots were developed using ClarityTM Western ECL substrate (Bio-Rad) and exposed on a ChemiDocTM Touch Imaging System (Bio-Rad).

### Dynamic scanning fluorimetry (DSF)

DSF on WT and mutated NS1 protein at a concentration of 100 μM was performed on a Tycho instrument (Nanotemper Technologies) at a scanning rate of 30°C/min in the 35–95°C temperature range. Intrinsic fluorescence of tryptophan (NS1 W90) was recorded at 350 and 330 nm.

### Bio-Layer Interferometry (BLI)

All proteins were dialyzed into 20 mM Tris pH 8.0, 200 mM NaCl, and 1 mM TCEP buffer. 0.05% Tween was added for BLI experiments. His-MED25 ACID (1 μM) was captured on Octet Ni-NTA biosensors (Sartorius) for 180 s at a level of 2.0 mm. Kinetic experiments were performed on an Octet RED96e system (FortéBio) at 25°C and under 1000 rpm shaking, using black 96-well plates. His-MED25 ACID-loaded biosensors were first equilibrated in buffer for 60 s (baseline), then incubated for 300 s in a concentration series of WT or mutated NS1 solutions, obtained by 2-fold dilutions (for association), and finally incubated for 600 s in buffer (for dissociation). Real-time binding kinetics were analyzed with the Octet Analysis Studio software (v.12.2.2.6). The raw signal was processed by subtracting two reference signals, the first measured from a biosensor without His-MED25 ACID and the second measured in the absence of NS1. A global fitting of both association and dissociation signals was done either with a 1:1 binding model or with a heterogeneous 2:1 model.

### Isothermal Titration Calorimetry (ITC)

ITC measurements were carried out on a MicroCal PEAQ-ITC calorimeter (Malvern Panalytical) at a temperature of 25°C. Protein samples (WT or mutated NS1, and MED25 ACID) were dialyzed against the same batch of 20 mM Tris pH 8.0, 200 mM NaCl, 1 mM TCEP buffer. NS1 (20 μM) was placed into the calorimeter sample cell (V = 200 μL). Aliquots of 2 μL of MED25 ACID at a concentration of 200 μM (for titration of WT NS1 and NS1Δα3) or 250 μM (for titration of NS1 with single amino acid substitutions) placed in the 40 μL syringe were injected into the NS1 protein solution under stirring at 500 rpm every 180 s during 4 s. Data were processed and analyzed with the MicroCal PEAQ-ITC Analysis Software (v1.41) according to the one set of sites binding model.

### Nuclear Magnetic Resonance (NMR)

2D $^1$H-$^{15}$N HSQC spectra of $^{15}$N-labeled MED25 ACID were measured on a Bruker 800 MHz NMR spectrometer equipped with a TCI cryoprobe, at a temperature of 20°C. Samples contained 100 μM $^{15}$N-MED25 ACID, with or without 0.5-1.0 molar equivalent of WT or mutated NS1 in 20 mM Na phosphate pH 6.5, 100 mM NaCl, 5 mM TCEP buffer. 7.5% D$_2$O was added to lock the spectrometer frequency. Spectra were processed with TopSpin 4.0 software (Bruker BioSpin) and analyzed with AnalysisAssign V3.1.1 [80] software.

### NanoLuc interaction assay

Constructs expressing the NanoLuc subunits 114 and 11S were used [47]. HEK 293T cells were seeded at a concentration of 6 x $10^4$ cells per well in a 48-well plate. After 24 h, cells were co-transfected in triplicate with 0.4 μg of total DNA (0.2 μg of each plasmid) using Lipofectamine 2000 (Invitrogen). 24 h post-transfection, cells were washed with PBS and lysed for 1 h at room temperature using 50 μL NanoLuc lysis buffer (Promega). NanoLuc enzymatic activity was measured using the Nano-Glo Substrate (Promega). For each pair of plasmids, normalized luminescence ratios (NLRs) were calculated as follows: the luminescence activity measured in cells transfected with the two plasmids (each viral protein fused to a different NanoLuc subunit) was divided by the sum of luminescence activities measured in two control samples obtained by co-transfecting one NanoLuc fused to viral or host protein plasmid with a plasmid expressing only the other NanoLuc subunit. Data represent the mean and standard deviation (SD) of 4 independent experiments, each done in triplicate. Luminescence was measured with an Infinite 200 PRO reader (TECAN).

## Immunofluorescence microscopy

Overnight cultures of BEAS-2B cells seeded at 5 x $10^5$ cells per well in 6-well plates (on an 18-mm micro cover glass for immunostaining) were transfected with 2 µg of pCI-neo plasmids expressing WT or mutated NS1 using Lipofectamine 2000 (Invitrogen). At 24 h post-infection, cells were fixed with 4% paraformaldehyde for 10 min, blocked with 3% BSA and 0.2% Triton X100–PBS for 10 min, and stained with a monoclonal rabbit NS1 antibody (GeneTex, GTX638591) (1:200), followed by rabbit secondary antibody conjugated to Alexa Fluor 488 (A11008, Invitrogen) (1:1000). Images were acquired using a white light laser SP8 (Leica Microsystems, Wetzlar, Germany) confocal microscope at a nominal magnification of 63 and the Leica Application Suite X (LAS X) software.

## RT-qPCR

Overnight cultures of BEAS-2B cells seeded at 2 x $10^5$ in a 24-well plate were infected with recombinant WT or mutated NS1 rRSV-mCherry, at an MOI of 3. At 10 h post-infection, the supernatant of the infected cells was removed, and the cells were washed with PBS and frozen at -80°C until RNA extraction. The RNA extraction was performed using the TRI Reagent (Molecular Research Center) according to the manufacturer's protocol. Total cDNA was synthesized from 69 ng of RNA using the iScript Advanced cDNA Synthesis Kit for RT-qPCR (Bio-Rad). 0.7 ng of cDNA template was added to 20 µL of reaction mixture containing 10 nM of primers for the ISGs OAS1A, IFITM3, ISG15, IFIT1, IFIT2, IFIT3, and Mx2 and iTaq Universal SYBR Green Supermix (Bio-Rad). Each point was performed in triplicate. The reaction was performed with a RealPleax2 thermocycler (Eppendorf). Fluorescence was monitored during the qPCR reaction. Viral RNA expression was quantified using the ΔΔCt method and normalized to the housekeeping gene GAPDH, β-actin, and 18S expression levels. Primers are listed in S3 Table.

## Quantification of interferons in supernatants

IFN-α, IFN-β, and IFN-λ protein levels were quantified in the supernatant of BEAS-2B cells infected with WT rRSV-mCherry or rRSV-mCherry with NS1 mutations at an MOI of 3 using the ProcartaPlex Human Basic Kit coupled with IFN-α, IFN-β, and IFN-λ ProcartaPlex Simplex Kit (Thermofisher) following the manufacturer's instructions. Quantification was performed on a MAGPIX system (Luminex), and data were analyzed with the Bio-Plex Manager software (Bio-Rad, Hercules, CA). The concentration of each IFN was determined from standard curves.

## siRNA transfection and infection

MED25 siRNA (sequence GCCCTTTGTTCCGGAACTCAA) (Qiagen) and a negative control siRNA (Qiagen 1027310) were used. A549 cells were transfected with siRNAs at a final concentration of 10 nM by reverse transfection in 24-well plates using Lipofectamine RNAiMAX (Thermo Fisher) according to the manufacturer's instructions. Briefly, a mixture containing Opti-MEM (Invitrogen), Lipofectamine RNAiMAX, and siRNA was incubated for 20 min at room temperature before being deposited at the bottom of the wells. The cells were then added dropwise before incubation at 37°C with 5% $CO_2$. 24 h post-transfection, the medium was removed, and the cells were infected either with rRSV-mCherry or with rRSV-mKate2 at an MOI of 0.5 in DMEM without phenol red and without FCS for 2 h at 37°C. The medium was then replaced by DMEM supplemented with 2% SVF, and the cells were incubated for 10, 24 and 48 h at 37°C. Quantification of replication was performed by measuring mCherry or mKate2 fluorescence (excitation at 580 nm and emission at 620 nm) using an Infinite M200 PRO (TECAN) luminometer. Noninfected A549 cells were used as standards for fluorescence background levels. Each experiment was performed in duplicate and repeated at least four times. For each experiment, cells treated under the same conditions were lysed, and MED25 expression was analyzed 72 h post-transfection by Western blotting on an immune-precipitated MED25 sample using a polyclonal rabbit MED25 antibody (Sigma-Aldrich, HPA068802) (1:5000).

## Toxicity assay

Cell viability was measured on A549 cells transfected with control or MED25 siRNA by using the CellTiter-Glo Assay (Promega). The CTG reagent was mixed with the supernatant of cells in a ratio of 1:1 per well and incubated for 2 min at room temperature under agitation. Luminescence quantification was performed using an Infinite M200 PRO reader (TECAN). Control wells containing medium without cells were used as standard luminescence background.

## Supporting information

**S1 Table:  Primers used for cloning and site-directed mutagenesis of RSV NS1, MED25 and TRIM25.**
(XLSX)

**S2 Table:  Primers used for reverse genetics.**
(XLSX)

**S3 Table:  Primers used for the ISG quantification by RT-qPCR.**
(XLSX)

**S1 Fig:  Comparison between theAlphaFold (AF) generated structural model and the X-ray crystallographic structure (PDB 9ccv) of the NS1–MED25 ACID complex.** NS1 is in blue, and MED25 ACID in white in the AF model. NS1 is in green cyan, and MED25 ACID in sand in the X-ray crystallographic structure. The RMSD calculated on 774 Cα atoms is 0.787 Å. The two views show the H1 and the H2 interfaces of MED25 ACID.
(TIF)

**S2 Fig:  Bio-layer interferometry real time association and dissociation curves of wild-type full-length or mutated NS1.** Measurements at 25°C were performed with bound His-tagged MED25 ACID at pH 8.0 and at different concentrations of NS1 proteins, as indicated below each graph. Fitted curves are in red lines, and fitting parameters are indicated in the upper right corner of each graph. The sum of squared deviations Chi$^2$ and the coefficient of determination R$^2$ are indicators for the quality of curve fits.
(TIF)

**S3 Fig:  Dynamic scanning fluorimetry data measured on wild-type and mutated NS1 at pH 8.** The ratio between fluorescence at 350 and 330 nm and the first derivative are represented as a function of temperature. The inflection temperature is reported in the bar diagram. The value for WT NS1 is a mean value obtained from 2 independent measurements, and the error bar represents the standard deviation.
(TIF)

**S4 Fig:  AlphaFold structural prediction of the complex between human MED25 ACID and RSV NS1Δα3. (A)** Proteins are in cartoon representation and colored according to the pLDDT (predicted local-distance difference test) confidence score, using the ChimeraX [43] AlphaFold color palette. The predicted aligned error (PAE) matrix for the complex was plotted with a color code from dark green to white representing the expected position error ranging from 0 to 30 Å. **(B)** The NS1Δα3–MED25 ACID (MED25 ACID in grey and NS1Δα3 in green) and NS1–MED25 ACID (MED25 ACID in light blue and NS1 in blue) complex models were structurally aligned for comparison.
(TIF)

**S5 Fig:  ITC binding isotherms for wild-type and mutated NS1 binding to MED25 ACID.** Measurements were carried out in 20 mM Tris pH 8.0, 200 mM NaCl, 1 mM TCEP and at a temperature of 25°C. For each variant, raw binding data are shown on top, and integrated titration curves at the bottom. The NS1 concentration in the calorimeter cell (V = 200 μL) was 20 μM. MED25 ACID at a concentration of 200 μM (for experiments with WT NS1 and NS1Δα3) or 250 μM (for NS1 with

single amino acid NS1 mutations) was injected in 2 µL volumes, under stirring at 500 rpm. The duration of injection was 4 s, separated by a delay time of 180 s.
(TIF)

**S6 Fig: Protein sequence alignments of RSV NS1 and M proteins from recombinant WT rRSV-mCherry and NS1 mutants.** Sequences were obtained from next-generation sequencing of the viral genomes. The alignment was performed with ClustalW and prepared with ESPript3. Residues in red are conserved residues.
(TIF)

**S7 Fig: Analysis of the expression of IFITM3 and IFIT1 ISGs in BEAS-2B cells by Western blotting.** BEAS-2B cells were infected with WT rRSV-mCherry and the most attenuated NS1 mutants at an MOI of 3. At 16 h pi, infected cells were lysed and subjected to a Western Blot analysis using IFITM3, IFIT1, and tubulin antibodies at 1:1000 dilution. IFIT1 protein levels were quantified using ImageJ software.
(TIF)

**S8 Fig: Impact of MED25 knockdown on rRSV in A549 cells.** A549 cells were transfected with 10 nM of control siRNA or MED25 siRNA, followed by infection 24 h later with **(A)** rRSV-mKate2 A2 strain or **(B)** WT and NS1 mutant rRSV-mCherry Long strain at an MOI of 0.5. RSV replication was quantified **(A)** at 48 h pi by measurement of mKate2 fluorescence intensity, or **(B)** at 10 h and 24 h pi by measurement of mCherry fluorescence intensity. Data show means and standard errors of 4 independent experiments in **(A)** and 3 independent experiments in **(B)**. $p < 0.05$ *, $p < 0.005$ **, $p < 0.0005$ *** (t-test).
(TIF)

## Acknowledgments

We thank Christine Lenoir for technical assistance with the production of recombinant proteins, and Marie Galloux for helpful discussions.

## Author contributions

**Conceptualization:** Monika Bajorek, Christina Sizun.

**Formal analysis:** Celia Ait-Mouhoub, Jiawei Dong, Magali Noiray, Alexis Verger, Delphyne Descamps, Monika Bajorek, Christina Sizun.

**Funding acquisition:** Magali Noiray, Delphyne Descamps, Monika Bajorek, Christina Sizun.

**Investigation:** Celia Ait-Mouhoub, Jiawei Dong, Magali Noiray, Jenna Fix, Stepanka Nedvedova, Slim Fourati, Alexis Verger, Jean-Francois Eleouet, Delphyne Descamps, Monika Bajorek, Christina Sizun.

**Methodology:** Magali Noiray, Alexis Verger, Delphyne Descamps, Monika Bajorek, Christina Sizun.

**Resources:** Magali Noiray, Jenna Fix, Slim Fourati, Alexis Verger, Jean-Francois Eleouet, Delphyne Descamps, Monika Bajorek, Christina Sizun.

**Supervision:** Delphyne Descamps, Monika Bajorek, Christina Sizun.

**Validation:** Celia Ait-Mouhoub, Jiawei Dong, Magali Noiray, Slim Fourati, Alexis Verger, Delphyne Descamps, Monika Bajorek, Christina Sizun.

**Visualization:** Celia Ait-Mouhoub, Magali Noiray, Monika Bajorek, Christina Sizun.

**Writing – original draft:** Celia Ait-Mouhoub, Monika Bajorek, Christina Sizun.

**Writing – review & editing:** Celia Ait-Mouhoub, Jiawei Dong, Magali Noiray, Jenna Fix, Stepanka Nedvedova, Slim Fourati, Alexis Verger, Jean-Francois Eleouet, Delphyne Descamps, Monika Bajorek, Christina Sizun.

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
