## [Decision Letter · Decision Letter 0]

26 Mar 2025

PPATHOGENS-D-25-00193

A dual interaction between RSV NS1 and MED25 ACID domain reshapes antiviral responses

PLOS Pathogens

Dear Dr. Sizun,

Thank you for submitting your manuscript to PLOS Pathogens. After careful consideration, we feel that it has merit but does not fully meet PLOS Pathogens's publication criteria as it currently stands. Therefore, we invite you to submit a revised version of the manuscript that addresses the points raised during the review process.

Please submit your revised manuscript within 60 days May 25 2025 11:59PM. If you will need more time than this to complete your revisions, please reply to this message or contact the journal office at plospathogens@plos.org. Please include the following items when submitting your revised manuscript:

We look forward to receiving your revised manuscript.

Kind regards,

Alexander Bukreyev, Ph.D.

Academic Editor

PLOS Pathogens

Matthias Schnell

Section Editor

PLOS Pathogens

 Sumita Bhaduri-McIntosh

Editor-in-Chief

PLOS Pathogens

orcid.org/0000-0003-2946-9497

 Michael Malim

Editor-in-Chief

PLOS Pathogens

orcid.org/0000-0002-7699-2064

**Journal Requirements:**

At this stage, the following Authors/Authors require contributions: Celia Ait-Mouhoub, Jiawei Dong, Magali Noiray, Jenna Fix, Stepanka Nedvedova, Alexis Verger, Jean-Francois Eleouet, Delphyne Descamps, Monika Bajorek, and Christina Sizun. Please ensure that the full contributions of each author are acknowledged in the "Add/Edit/Remove Authors" section of our submission form.

3) Please ensure that the funders and grant numbers match between the Financial Disclosure field and the Funding Information tab in your submission form. Note that the funders must be provided in the same order in both places as well. Currently, the order of the funders is different in both places.

**Reviewers' Comments:**

Reviewer's Responses to Questions

**Part I - Summary**

Reviewer #1: Strengths:

- new insights in the RSV NS1-MED25 interaction: dual mode interaction involving two interfaces

- Multiple assays (BLI, ICT, NMR, split luciferase complementation assay result in ave very similar conclusion

- Biochemical studies are corroborated by experiments that were performed in RSV-permissive mammalian cells.

Reviewer #2: This manuscript studies the importance of an interaction between a virus protein, NS1 of Respiratory Syncytial Virus (RSV), that is known to orchestrate IFN antagonism, and a host MED25 protein, which is one of the subunits of the mediator complex. This interaction was known to be key to modulate host transcription in nucleus, a process that is hypothesized to be important for its IFN antagonism. They computationally predict and show that the NS1 C-terminal region interacts with MED25 via the H2 interface of the transcription factor activating TAD domain. They identify key residues in NS1 that are important for this interaction and show that mutating these attenuated RSV replication and induced robust IFN signaling. Despite the thorough biophysical work for predicting and showing protein-protein interaction and identify key NS1 residues for their interaction, the manuscript fails to comprehensively show that these mutations induced higher IFN signaling. Even though the RSV mutants made has a profound replication defect, there is not enough evidence to show the mutants induced IFN signaling as there is no direct measure of increased IFN production. Despite showing increased ISG production via qPCR, the ISGs protein levels should also be quantified to comprehensively show ISG stimulation. Most importantly, a clear link between MED25 and IFN signaling was not established, exacerbated by a replication phenotype in contrast with the previous study of MED25 KO A549 cells. Hence for all these reasons, the manuscript should address the major concerns addressed to be considered for publication.

MAJOR CONCERN

1) It is increasingly becoming more important to show that the virus mutants made via reverse genetics approaches are confirmed for lack of off-target polar mutations both at a plasmid-levels and at a virus level. The observed phenotypes could be dues to random mutations elsewhere in the virus genome and hence validation of these plasmid and virus sequences need to be submitted to ensure that the observed phenotypes were indeed due to the mutations introduced.

2) Whenever dealing with interaction of proteins at specific domains, it is necessary to include line-diagrams for those proteins showing basic linear domain architecture, so that readers could easily follow. This manuscript will heavily benefit from showing where the intended domains were present in NS1 and MED25 using simple cartoons.

3) Whenever showing IFN stimulation, it is necessary to directly quantify IFN levels before showing ISG stimulation. Hence a qPCR showing IFNα/β levels and IFNλ levels (IFNγ are minimally expressed in the cells used and hence it is not necessary to show that) is necessary before showing ISG stimulation.

4) Along with transcript quantification, it is also important to show protein levels of IFNs and ISGs, as eventually protein level differences are what lead to replication defects and other phenotypes. Hence ELISA or a western blot of the IFNs and ISGs are also important to make these claims.

5) It is confusing to see a decrease in WT virus in MED25KO cells and less of a defect in corresponding NS1 mutants. This could be a virus replication issue and could be caused due to temporal changes. Hence, it is important to show replication of WT and the corresponding RSV mutants atleast in 3 different timepoints (early, mid and late replication timepoint) in A549 cells, as it may help in seeing better differences in replications of the mutants compared to WT.

MINOR CONCERNS

1) Line 37 – “IFIT1”

2) Line 93 – please replace the word “transduce” with another word as transduce has a different meaning in virology.

3) Line 102 – A possible link

4) Line 338-339 – In this context, Δα3 is used as a negative control as there is a profound replication defect expected. But this is debatable and hence is left to the authors discretion

5) Line 405-406 - HRP instead of HPR

**Part II – Major Issues: Key Experiments Required for Acceptance**

Reviewer #1: The MED25 knock down experiments were performed in A549 cells with the RSV-mCherry reporter viruses, as were all other experiments with RSV infection. The authors should also assess the effect of MED25 knock down on the replication of a lab strain of RSV A (or B) or, if available, a clinical isolate of RSV to confirm the observation that RSV replication is restricted in MED25 knock down cells.

Reviewer #2: 1) It is increasingly becoming more important to show that the virus mutants made via reverse genetics approaches are confirmed for lack of off-target polar mutations both at a plasmid-levels and at a virus level. The observed phenotypes could be dues to random mutations elsewhere in the virus genome and hence validation of these plasmid and virus sequences need to be submitted to ensure that the observed phenotypes were indeed due to the mutations introduced.

2) Whenever dealing with interaction of proteins at specific domains, it is necessary to include line-diagrams for those proteins showing basic linear domain architecture, so that readers could easily follow. This manuscript will heavily benefit from showing where the intended domains were present in NS1 and MED25 using simple cartoons.

3) Whenever showing IFN stimulation, it is necessary to directly quantify IFN levels before showing ISG stimulation. Hence a qPCR showing IFNα/β levels and IFNλ levels (IFNγ are minimally expressed in the cells used and hence it is not necessary to show that) is necessary before showing ISG stimulation.

4) Along with transcript quantification, it is also important to show protein levels of IFNs and ISGs, as eventually protein level differences are what lead to replication defects and other phenotypes. Hence ELISA or a western blot of the IFNs and ISGs are also important to make these claims.

5) It is confusing to see a decrease in WT virus in MED25KO cells and less of a defect in corresponding NS1 mutants. This could be a virus replication issue and could be caused due to temporal changes. Hence, it is important to show replication of WT and the corresponding RSV mutants atleast in 3 different timepoints (early, mid and late replication timepoint) in A549 cells, as it may help in seeing better differences in replications of the mutants compared to WT.

**Part III – Minor Issues: Editorial and Data Presentation Modifications**

Reviewer #1: 1. Lines 405-406: HPR > HRP.

2. Lines 454 and 487: “data are from 4 independent experiments”. However, for most of the RSV mutants there are 5 or more data points depicted in the graph. Please clarify.

3. Line 490: mortality > cell death.

4.. Line 662: please specify the source of the A549 cells.

5. Line 670: please specify in which figure results on infection with RSV A Long are presented.

6. Please clarify whether the rescued RSVs were sequence verified.

7. Line 752: please clarify the statement.

Reviewer #2: 1) Line 37 – “IFIT1”

2) Line 93 – please replace the word “transduce” with another word as transduce has a different meaning in virology.

3) Line 102 – A possible link

4) Line 338-339 – In this context, Δα3 is used as a negative control as there is a profound replication defect expected. But this is debatable and hence is left to the authors discretion

5) Line 405-406 - HRP instead of HPR

PLOS authors have the option to publish the peer review history of their article (what does this mean? ). If published, this will include your full peer review and any attached files.

**Do you want your identity to be public for this peer review?** For information about this choice, including consent withdrawal, please see our Privacy Policy .

Reviewer #1: No

Reviewer #2: No

**Figure resubmission:**
---

## [Decision Letter · Decision Letter 1]

21 Jul 2025

Dear Dr Sizun,

We are pleased to inform you that your manuscript 'A dual interaction between RSV NS1 and MED25 ACID domain reshapes antiviral responses' has been provisionally accepted for publication in PLOS Pathogens.

Best regards,

Matthias Johannes Schnell, PhD

Section Editor

PLOS Pathogens

Matthias Schnell

Section Editor

PLOS Pathogens

Sumita Bhaduri-McIntosh

Editor-in-Chief

PLOS Pathogens

orcid.org/0000-0003-2946-9497

Michael Malim

Editor-in-Chief

PLOS Pathogens

orcid.org/0000-0002-7699-2064

Reviewer Comments (if any, and for reference):

Reviewer's Responses to Questions

**Part I - Summary**

Reviewer #1: (No Response)

Reviewer #2: This manuscript studies the importance of an interaction between a virus protein, NS1 of Respiratory Syncytial Virus (RSV), that is known to orchestrate IFN antagonism, and a host MED25 protein, which is one of the subunits of the mediator complex. This interaction was known to be key to modulate host transcription in nucleus, a process that is hypothesized to be important for its IFN antagonism. They computationally predict and show that the NS1 C-terminal region interacts with MED25 via the H2 interface of the transcription factor activating TAD domain. They identify key residues in NS1 that are important for this interaction and show that mutating these attenuated RSV replications and induced robust IFN signaling. All the previously outlined major and minor reviews have been thoroughly addresses and is now in a better position to be published.

**Part II – Major Issues: Key Experiments Required for Acceptance**

Reviewer #1: (No Response)

Reviewer #2: None

**Part III – Minor Issues: Editorial and Data Presentation Modifications**

Reviewer #1: (No Response)

Reviewer #2: None

PLOS authors have the option to publish the peer review history of their article (what does this mean? ). If published, this will include your full peer review and any attached files.

**Do you want your identity to be public for this peer review?** For information about this choice, including consent withdrawal, please see our Privacy Policy .

Reviewer #1: No

Reviewer #2: No

---

## [Editor Report · Acceptance letter]

Dear Dr Sizun,

We are delighted to inform you that your manuscript, " 

A dual interaction between RSV NS1 and MED25 ACID domain reshapes antiviral responses," has been formally accepted for publication in PLOS Pathogens.

Best regards,

Sumita Bhaduri-McIntosh

Editor-in-Chief

PLOS Pathogens

orcid.org/0000-0003-2946-9497

Michael Malim

Editor-in-Chief

PLOS Pathogens

orcid.org/0000-0002-7699-2064